# Meta Flow Maps Enable Scalable Reward Alignment

Peter Potaptchik [* 1 2]  Adhi Saravanan [* 1]  Abbas Mammadov [1]  Alvaro Prat [1]
Michael Albergo [† 3 2]  Yee Whye Teh [† 1]

## Abstract

Controlling generative models—whether via inference-time steering or fine-tuning—is expensive. Control relies on estimating the value function—typically necessitating costly trajectory simulations. To eliminate this bottleneck, we introduce **Meta Flow Maps (MFMs)**, stochastic extensions of consistency models and flow maps. MFMs are trained to perform **one-step posterior sampling**, generating arbitrarily many i.i.d. draws of clean data $x_1$ from any noisy state $x_t$. Crucially, these samples are differentiable in the conditioning state $x_t$, unlocking efficient estimation of the value function gradient. We leverage this capability to enable both *inference-time steering* without inner rollouts, and unbiased, off-policy *fine-tuning* to general rewards. Among our fine-tuning and steering experiments on ImageNet, we highlight that our single-particle steered-MFM sampler outperforms a Best-of-1000 baseline across multiple rewards at a fraction of the compute.

🌐 Project Page  ⭘ Code

## 1. Introduction

Many of the most powerful generative models are transport-based, as in diffusion models, flow matching, and stochastic interpolants (Song et al., 2020; Ho et al., 2020; Lipman et al., 2022; Albergo & Vanden-Eijnden, 2022). A growing frontier is adapting these models to align with a reward function. In many applications, we are not satisfied with unconditional samples; instead, we want to *control* model trajectories so that samples exhibit desirable properties (Bansal et al., 2023; Kim et al., 2023; Ma et al., 2025).

Formally, this task can be phrased as sampling from a modi-

---
[*]Equal contribution [†]Senior authors. [1]Department of Statistics, University of Oxford [2]Kempner Institute [3]Harvard University. Correspondence to: Peter Potaptchik <peter.potaptchik@stats.ox.ac.uk>, Adhi Saravanan <adhithya.saravanan@stats.ox.ac.uk>.

*Proceedings of the 43$^{rd}$ International Conference on Machine Learning*, Seoul, South Korea. PMLR 306, 2026. Copyright 2026 by the author(s).

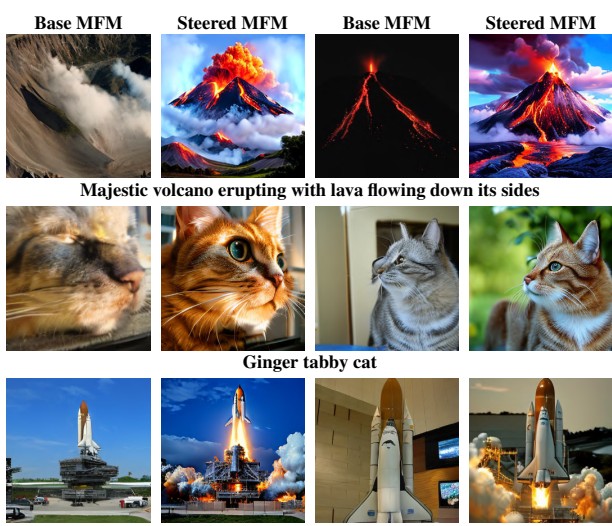

**Base MFM**  **Steered MFM**  **Base MFM**  **Steered MFM**

Majestic volcano erupting with lava flowing down its sides

Ginger tabby cat

Space shuttle launching into space, fiery fumes

*Figure 1.* Inference-time MFM steering on HPSv2 using the prompts shown. Base MFM uses **only class labels** for sampling. We initialize both trajectories from the same noise realization.

fied target, the *reward-tilted* distribution:

$$p_{\text{reward}}(x) \propto p_{\text{model}}(x) e^{r(x)}, \tag{1}$$

where $p_{\text{model}}(x)$ is the output distribution of the original pretrained generative model, and $r(x)$ is a general reward function. This formulation encapsulates settings such as classifier guidance, where $r(x) = \log p(c|x)$ targets a class $c$ (Dhariwal & Nichol, 2021), or black-box rewards (e.g., $r$ measures aesthetics or prompt alignment). In all cases, the central algorithmic challenge is to modify the sampling dynamics so that terminal samples are distributed as $p_{\text{reward}}$.

Two related paradigms for control are *inference-time steering* and *fine-tuning*. Inference-time steering keeps the pretrained model fixed and modifies the sampling process directly—traditionally at the cost of substantially more expensive generation—in order to target $p_{\text{reward}}$. Fine-tuning, conversely, updates the generative network's parameters to permanently target $p_{\text{reward}}$. While effective for a single fixed objective, repeating this computationally intensive process for every new downstream reward is often intractable.

Both paradigms share a unified theoretical objective. Specifically, the optimal drift targeting $p_{\text{reward}}$ is identical in both

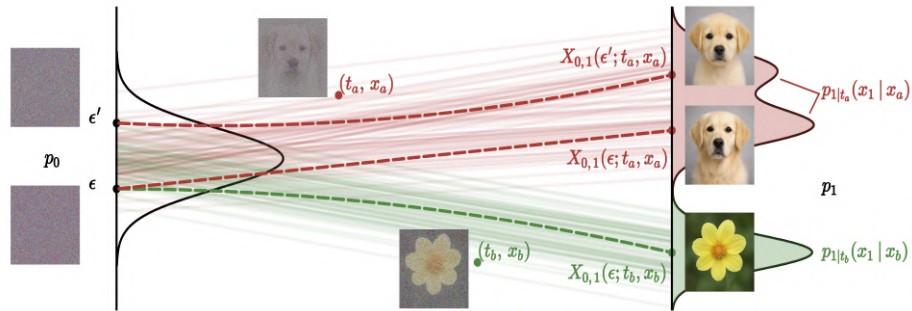

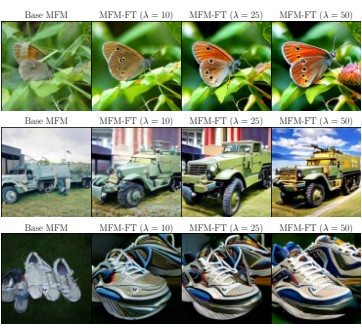

*Figure 2.* A MFM $X$ conditions on an intermediate time–state pair $(t, x)$ (noisy images) along the interpolant and learns a shared conditional flow $X_{s,u}(\cdot\,; t, x)$ that maps base noise $\epsilon \sim p_0$ to endpoint samples $x_1 \sim p_{1|t}(\cdot|x)$ (clean images) via $X_{0,1}(\epsilon; t, x)$. For a given $(t, x)$ pair, varying the initial noise $\epsilon' \sim p_0$ yields different samples from the same posterior $p_{1|t}(\cdot|x)$. Conversely, for the same initial noise $\epsilon$, conditioning on two different time–state pairs $(t_a, x_a)$ and $(t_b, x_b)$ yields samples from different posteriors, $p_{1|t_a}(\cdot|x_a)$ and $p_{1|t_b}(\cdot|x_b)$.

*Figure 3.* Samples from the class-conditioned ODE before and after MFM fine-tuning for different reward multipliers $\lambda$. We provide a larger set of samples in Figure 21.

cases, and can be expressed in terms of the value function which, for a stochastic process $(I_t)_{t\in[0,1]}$, is defined as

$$V_t(x) := \log \mathbb{E}[e^{r(I_1)} \mid I_t = x]. \quad (2)$$

This function measures the expected future reward, where the expectation is taken over the *conditional posterior* $p_{1|t}(\cdot|x)$—the distribution of clean endpoints $I_1$ consistent with the current state $I_t = x$. Crucially, the gradient $\nabla_x V_t(x)$ determines the optimal drift correction required to target $p_{\text{reward}}$ (Dai Pra, 1991). This correction can be applied transiently during steering or distilled permanently during fine-tuning. Thus, both paradigms are unified by the shared bottleneck of estimating $\nabla_x V_t(x)$ which generally necessitates samples from the conditional posterior $p_{1|t}(\cdot|x)$.

This creates a computational dilemma. Heuristics that approximate the posterior (Song et al., 2023a; Chung et al., 2024) are efficient but highly biased, often failing in multimodal settings. Conversely, Monte Carlo estimation requires drawing exact samples by integrating long ODE or SDE trajectories (Holderrieth et al., 2025; Jain et al., 2025). This reliance on repeated trajectory rollouts makes steering prohibitively slow and fine-tuning for every downstream reward impractical. To eliminate this bottleneck, we propose to compress these expensive rollouts into efficient, few-step maps during training—thereby making it feasible to efficiently steer or fine-tune on any new reward.

A natural strategy is to learn a new class of few-step maps, paralleling the development of consistency models and flow maps (Song et al., 2023b; Geng et al., 2025a; Boffi et al., 2025a). Flow maps distill ODE trajectories $(x_t)_{t\in[0,1]}$ into an efficient map that predicts the trajectory endpoint $x_1$ from an intermediate state $x_t$. However, while a flow map can target a specific posterior, as *deterministic* maps of $x_t$, they are inherently incapable of representing the diversity of the entire family of conditional posteriors $p_{1|t}(\cdot|x)$ simultaneously. We instead seek *stochastic* maps that efficiently compress generative rollouts while explicitly preserving the *full* posterior distribution.

We refer to this new class of operators as *Stochastic Flow Maps*. Concretely, we define a stochastic flow map as a transformation that maps an exogenous noise source $\epsilon$ and an intermediate state $x$ directly to a sample $x_1 \sim p_{1|t}(\cdot|x)$. By varying this noise $\epsilon$, the map can generate arbitrarily many i.i.d. draws from the posterior in a single step. These one-shot samples also provide a *differentiable reparametrization* of the posteriors, enabling asymptotically exact estimation of the value function gradient. This raises the natural question: how can we train such efficient stochastic flow maps?

To address this, we introduce **Meta Flow Maps (MFMs)**. We exploit that for any intermediate state $x$ at time $t$, there exists an auxiliary probability flow ODE transporting a simple noise distribution to the conditional posterior $p_{1|t}(\cdot|x)$. Since each ODE has a flow map compressing its trajectory rollouts, we reduce learning a stochastic flow map to learning this infinite collection of posterior-targeting flow maps. MFMs achieve this by training a single amortized model that acts as a "meta" flow map over this infinite family.

> **Core Contributions:**
> - We introduce Meta Flow Maps (MFMs), stochastic extensions of flow maps that generate **arbitrarily many one-step** samples from the posterior distribution of data $x_1$ conditioned on a noisy state $x_t$.
> - We leverage MFMs for efficient, asymptotically exact **inference-time steering** via Monte Carlo estimators of the value function gradient, while also presenting scalable MFM-enabled search methods.
> - We show that MFMs enable efficient **off-policy fine-tuning** using unbiased objectives.

## 2. Dynamical Measure Transport

**Dynamical Transport via ODEs.** Generative flow models aim to learn a transport that transforms samples from a reference distribution $x_0 \sim p_0$ into a data distribution $x_1 \sim p_1$, using existing data samples $\{x_1^{(i)}\}_{i=1}^N$. To this end, one constructs a drift $b_t : \mathbb{R}^d \to \mathbb{R}^d$ which defines an ODE:

$$\dot{x}_t = b_t(x_t), \qquad x_0 \sim p_0. \qquad (3)$$

**Training.** To ensure $x_1 \sim p_1$, flow matching and stochastic interpolants (Albergo & Vanden-Eijnden, 2022; Lipman et al., 2022; Liu et al., 2022) define a continuous-time stochastic process $(I_t)_{t \in [0,1]}$ which interpolates between samples from the prior $I_0 \sim p_0$ and the data $I_1 \sim p_1$ via

$$I_t = \alpha_t I_0 + \beta_t I_1, \qquad (4)$$

where $\alpha_t, \beta_t$ are time-dependent coefficients satisfying $\alpha_0 = \beta_1 = 1$ and $\alpha_1 = \beta_0 = 0$. The density $p_t$ of $I_t$ defines a continuous family of intermediate distributions bridging $p_0$ and $p_1$. One valid drift in (3) that generates this marginal family $(p_t)_{t \in [0,1]}$ (i.e., such that the density of $x_t$ is $p_t$) is given by the conditional expectation $b_t(x) = \mathbb{E}[\dot{I}_t | I_t = x]$ where $\dot{I}_t = \dot{\alpha}_t I_0 + \dot{\beta}_t I_1$ is the time derivative of the interpolant. In practice, $b_t$ is parameterized by a neural network $\hat{b}_t$ and learned by minimizing a regression loss:

$$b_t = \arg\min_{\hat{b}_t} \int_0^1 \mathbb{E} \left\| \hat{b}_t(I_t) - \dot{I}_t \right\|^2 dt. \qquad (5)$$

**One and Few-Step Sampling.** Numerically integrating the ODE in (3) requires many neural network evaluations, making inference expensive. This has motivated a class of methods including consistency models (Song et al., 2023b) and flow maps (Kim et al., 2024a; Frans et al., 2025; Boffi et al., 2025b; Sabour et al., 2025) that learn to directly predict the state of the ODE trajectory (3) at time $u$ from its state at time $s$. Concretely, a *flow map* $X_{s,u} : \mathbb{R}^d \to \mathbb{R}^d$ satisfies

$$X_{s,u}(x_s) = x_u, \quad \forall s, u \in [0,1], \qquad (6)$$

for trajectories $(x_t)_{t \in [0,1]}$ of the ODE. We parametrize the flow map as $X_{s,u}(x) = x + (u - s) v_{s,u}(x)$, where $v_{s,u}$ represents the *average velocity* of the trajectory between times $s$ and $u$. For $X_{s,u}$ to be consistent with the underlying ODE, the average drift must recover the instantaneous drift in the infinitesimal limit (Kim et al., 2024b):

$$\lim_{s \to u} \partial_u X_{s,u}(x) = v_{u,u}(x) = b_u(x). \qquad (7)$$

We learn a neural parameterization $\hat{v}_{s,u}$ with its induced map $\hat{X}_{s,u}$, and enforce the tangent condition along the diagonal $s = u$ via a flow matching objective as in (5):

$$\mathcal{L}_{\text{diag}}(\hat{v}) = \int_0^1 \mathbb{E} \left\| \hat{v}_{u,u}(I_u) - \dot{I}_u \right\|^2 du, \qquad (8)$$

where $(I_u, \dot{I}_u)$ is drawn from (4). To ensure $\hat{v}_{s,u}$ represents the average drift for $s \neq u$, it is equivalent to enforce the

semi-group property (Boffi et al., 2025b) for all $x \in \mathbb{R}^d$:

$$\hat{X}_{w,u}\big(\hat{X}_{s,w}(x)\big) = \hat{X}_{s,u}(x), \quad \forall s, w, u \in [0,1]. \qquad (9)$$

Existing methods therefore introduce an additional *consistency objective*, implicitly enforcing this condition and guaranteeing that $\hat{X}$ defines a valid flow map (Appendix F.1).

## 3. Reward Alignment

**Optimally Controlled Dynamics.** As discussed in Section 1, both inference-time steering and fine-tuning share a unified objective: sampling from the *reward-tilted distribution* $p_{\text{reward}}$. This requires modifying the base drift $b_t$ to an optimally controlled drift $b_t^\star$. Optimal control theory (Dai Pra, 1991) dictates that this correction is determined by the gradient of the value function $V_t(x)$ defined in (2). Specifically, letting $\frac{\sigma_t^2}{2} = \frac{\dot{\beta}_t}{\beta_t} \alpha_t^2 - \dot{\alpha}_t \alpha_t$, the optimally controlled ODE, generating samples $x_1^\star \sim p_{\text{reward}}$, is:

$$\dot{x}_t^\star = \underbrace{b_t(x_t^\star) + \frac{\sigma_t^2}{2} \nabla V_t(x_t^\star)}_{b_t^\star(x_t^\star)}, \quad x_0^\star \sim p_0. \qquad (10)$$

See Appendix B.1 for a derivation. Since the base drift $b_t$ is available from the pretrained model, the central algorithmic challenge lies in efficiently estimating $\nabla_x V_t(x)$.

**Estimating $\nabla_x V_t(x)$.** Estimating the value function in (2) necessitates sampling from the conditional posterior $p_{1|t}(\cdot | x)$. To do so, we seek a generative map $\Phi(\epsilon; t, x)$ that directly transforms noise $\epsilon \sim q$ into posterior samples, i.e., $\Phi(\epsilon; t, x) \sim p_{1|t}(\cdot | x)$. This allows us to express:

$$V_t(x) = \log \mathbb{E}_{\epsilon \sim q} \left[ e^{r(\Phi(\epsilon; t, x))} \right]. \qquad (11)$$

Crucially, because we require the *gradient* $\nabla_x V_t(x)$, we seek a *differentiable reparametrization* (Kingma & Welling, 2022) of the family of posterior distributions, that is, we require that $\Phi$ be differentiable in $x$. In particular, this will allow us to exchange the expectation with respect to $\epsilon$ with the gradient. This motivates the following definition.

> **Definition (Stochastic Flow Map).** A Stochastic Flow Map is a parametric function $\Phi(\epsilon; t, x) : \mathbb{R}^d \times [0,1] \times \mathbb{R}^d \to \mathbb{R}^d$ that maps exogenous noise $\epsilon \sim q$ directly to the target posterior $p_{1|t}(\cdot | x)$. Concretely, it satisfies the *conditional* transport constraint:
>
> $$\Phi(\cdot; t, x) \# q = p_{1|t}(\cdot | x), \quad \forall t, x. \qquad (12)$$
>
> We furthermore require that $\Phi$ be differentiable in $x$.

Access to such a stochastic flow map $\Phi$ allows us to represent the value function gradient as:

$$\nabla_x V_t(x) = \nabla_x \log \mathbb{E}_{\epsilon \sim q} \left( e^{r(\Phi(\epsilon; t, x))} \right). \qquad (13)$$

From this representation, we derive the following estimator.

**Gradient-Based Estimator (MFM-G).** A consistent Monte Carlo estimator of $\nabla_x V_t(x)$, where $\epsilon^{(i)} \overset{\text{iid}}{\sim} q$, is

$$\widehat{\nabla_x V_t(x)} = \nabla_x \log \left( \frac{1}{N} \sum_{i=1}^{N} e^{r(\Phi(\epsilon^{(i)}; t, x))} \right). \quad (14)$$

Since this estimation must be performed at every step of generation during steering, or at every iteration of training during fine-tuning, the practical viability of this approach hinges entirely on the efficiency of the stochastic flow map $\Phi$. Traditional posterior sampling strategies—which typically rely on expensive iterative rollouts of auxiliary ODEs or SDEs—are computationally inadequate for this task, as differentiating through long trajectories is prohibitively expensive (Appendix D). This leads to our core problem:

**Problem.** How can we train stochastic flow maps?

# 4. Meta Flow Maps

To address this, we introduce Meta Flow Maps. Intuitively, for every context $(t, x)$, we construct a specific conditional auxiliary ODE that transports the prior to the posterior $p_{1|t}(\cdot|x)$. We then train a single unified model to amortize the solution operators of this infinite family into accessible few-step maps.

## 4.1. Meta Flow Map Framework

**Conditional Probability Flow ODEs.** Our framework relies on the existence of a *conditional auxiliary ODE* defined for every context $(t, x) \in [0, 1] \times \mathbb{R}^d$. We seek a drift $\bar{b}_s(\cdot; t, x)$ that defines a flow transporting the prior $p_0$ directly to the posterior $p_{1|t}(\cdot|x)$. Theoretically, this drift is the solution to a conditional flow matching problem where the target distribution $p_1$ is replaced by the specific posterior $p_{1|t}(\cdot|x)$. Concretely, for $\bar{I}_0 \sim p_0$ and $\bar{I}_1 \sim p_{1|t}(\cdot|x)$, this drift is defined as:

$$\bar{b}_s(\bar{x}; t, x) = \mathbb{E}[\tfrac{d}{ds}\bar{I}_s | \bar{I}_s = \bar{x}], \quad \bar{I}_s = \alpha_s \bar{I}_0 + \beta_s \bar{I}_1. \quad (15)$$

The resulting conditional auxiliary ODE satisfies:

$$\frac{d}{ds}\bar{x}_s = \bar{b}_s(\bar{x}_s; t, x), \;\; \bar{x}_0 \sim p_0 \implies \bar{x}_1 \sim p_{1|t}(\cdot|x). \quad (16)$$

**Parametrizing the Family of Flows.** Associated with this collection of ODEs is an infinite *family* of flow maps, where each map acts as the solution operator for the ODE targeting the posterior $p_{1|t}(\cdot|x)$. We introduce the Meta Flow Map as a single map that unifies this entire family. By taking the context $(t, x)$ as an input, it indexes into this infinite family of solution operators, thereby realizing a "meta" flow map.

**Definition (Meta Flow Map).** A Meta Flow Map (MFM) is a map $X_{s,u}(\cdot; t, x) : \mathbb{R}^d \to \mathbb{R}^d$ which, for any context $(t, x)$, yields the solution to the conditional ODE in (16). Formally, for a trajectory $(\bar{x}_\tau)_{\tau \in [0,1]}$ defined by the drift $\bar{b}_\tau(\cdot; t, x)$, it satisfies:

$$X_{s,u}(\bar{x}_s; t, x) = \bar{x}_u, \quad \forall s, u \in [0, 1]. \quad (17)$$

By setting the base noise $q = p_0$, and $\Phi(\epsilon; t, x) := X_{0,1}(\epsilon; t, x)$, the MFM satisfies the definition of a Stochastic Flow Map (12).

## 4.2. Training

We parametrize the MFM $\hat{X}$ in residual form

$$\hat{X}_{s,u}(\bar{x}; t, x) = \bar{x} + (u - s)\hat{v}_{s,u}(\bar{x}; t, x), \quad (18)$$

where $\hat{v}$ is a neural network that predicts the average velocity of the auxiliary trajectory. While standard flow map training targets a single distribution, our goal is to learn a continuum of such maps in one model. For any *fixed* context $(t, x)$, the subnetwork $\hat{X}_{s,u}(\cdot; t, x)$ constitutes the specific flow map targeting $p_{1|t}(\cdot|x)$. We train this unified model by sampling contexts $(t, x)$, ensuring the network minimizes the loss in expectation over all these tasks. As with standard flow map training, we employ a *diagonal loss*, $\mathcal{L}_{\text{diag}}$, to anchor the instantaneous velocity field $\hat{v}_{s,s}(\cdot; t, x)$ to the ground-truth velocity $\bar{b}_s(\cdot; t, x)$ of the auxiliary conditional ODE. We use an additional *consistency loss*, $\mathcal{L}_{\text{cons}}$, to ensure the map $\hat{X}_{s,u}(\cdot; t, x)$ integrates this instantaneous velocity correctly over time intervals $(s, u)$, resulting in a valid flow map for each fixed $(t, x)$. We present two distinct regimes for computing these losses: training from data, and distillation from a pretrained model.

### 4.2.1. TRAINING FROM DATA

When training from scratch on a dataset, we construct targets using coupled interpolants and use a self-consistency loss.

**Diagonal Loss.** To ensure $\bar{v}_{s,s}$ learns the auxiliary drift $\bar{b}_s$, we construct reference trajectories that connect the prior to the correct conditional posterior. To do so, we sample times $s, t \sim \text{Unif}[0, 1]$, draw data $I_1 \sim p_1$, set $\bar{I}_1 = I_1$, draw independent $I_0, \bar{I}_0 \sim p_0$, and construct coupled interpolants:

$$I_t = \alpha_t I_0 + \beta_t I_1, \qquad \bar{I}_s = \alpha_s \bar{I}_0 + \beta_s \bar{I}_1. \quad (19)$$

Conditioned on the context state $I_t = x$, the endpoint $I_1$ is distributed exactly according to $p_{1|t}(\cdot|x)$, implying that Law $(\bar{I}_0, \bar{I}_1 \mid I_t = x) = p_0 \times p_{1|t}(\cdot|x)$. Therefore, the auxiliary path $\bar{I}_s$ traces a valid conditional trajectory from $p_0$ to this posterior. The time derivative, $\frac{d}{ds}\bar{I}_s = \dot{\alpha}_s \bar{I}_0 + \dot{\beta}_s \bar{I}_1$, serves as the regression target for the diagonal loss:

$$\mathcal{L}_{\text{diag}}^{\text{data}}(\hat{v}) := \int_0^1 \int_0^1 \mathbb{E} \left\| \hat{v}_{s,s}(\bar{I}_s; t, I_t) - \tfrac{d}{ds}\bar{I}_s \right\|^2 ds\, dt. \quad (20)$$

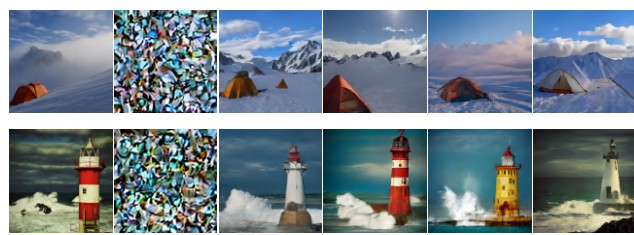 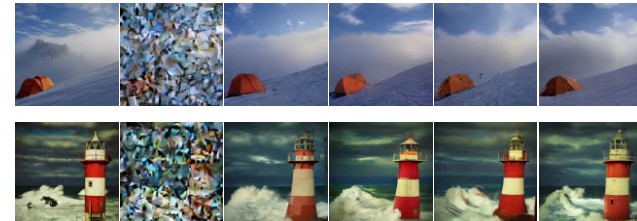

*(a)* **Large** noise level ($t = 0.2$). Posterior is broad and samples exhibit noticeable variation while maintaining semantic consistency.

*(b)* **Small** noise level ($t = 0.4$). Posterior concentrates and the conditional samples are highly similar to the original image.

*Figure 4.* **Conditional endpoint samples on ImageNet.** In each subfigure, the first column shows a data point from ImageNet, the second column shows a corrupted version $x_t$ at the indicated noise level, and the remaining four columns are four one-shot samples from the Meta Flow Map, $\hat{x}_1^{(i)} = X_{0,1}(\epsilon^{(i)}; t, x_t)$, targeting $p_{1|t}(\cdot|x_t)$, for independent $\epsilon^{(i)}$. The noise variables $\epsilon^{(i)}$ for the posterior-sample columns are coupled across the left and right subfigures, showing how the same $\epsilon^{(i)}$ yields different endpoints as $(t, x_t)$ changes.

By (15), minimizing the diagonal loss enforces that $\hat{v}_{s,s}(\bar{x}; t, x) = \bar{b}_s(\bar{x}; t, x)$ for all $t \in [0, 1], \bar{x}, x \in \mathbb{R}^d$.

**Consistency Loss.** To ensure that the learned map $\hat{X}_{s,u}(\cdot; t, x)$ constitutes a valid flow for any fixed $(t, x)$, we must enforce the consistency rules in (9). Consequently, we can employ any standard *self-consistency* loss (Boffi et al., 2025a) on the conditional map $\hat{X}_{s,u}(\cdot; t, x)$ holding $(t, x)$ fixed. Table 2 details four representative examples, though we emphasize that our framework is agnostic to this specific choice.

### 4.2.2. DISTILLATION FROM A TEACHER MODEL

When the prior $p_0$ is Gaussian, we can distill an MFM directly from a pretrained unconditional flow matching model with drift $b_t$. This provides lower-variance supervision than data-driven training.

**Analytical Conditional Drift via GLASS.** This approach is enabled by the availability of the ground-truth conditional drift $\bar{b}_s$ in closed form. As derived in GLASS flows (Holderrieth et al., 2025), the conditional drift $\bar{b}_s(\cdot; t, x)$ can be computed analytically from the teacher's unconditional drift $b_t$ via:

$$\bar{b}_s(\bar{x}_s; t, x) = w_1 \bar{x}_s + w_2 b_{t^*}(S(\bar{x}_s, x)) + w_3 x, \quad (21)$$

where $w_1, w_2, w_3$ are scalars, $t^*$ is a reparametrized time, and $S$ is a linear term. We provide explicit expressions for these quantities in Appendix I. This expression acts as a stable, low-variance "teacher" for the MFM.

**Diagonal Loss.** We regress the MFM's instantaneous velocity directly onto this analytical target:

$$\mathcal{L}_{\text{diag}}^{\text{teach}}(\hat{v}) := \int_0^1 \int_0^1 \mathbb{E} \left\| \hat{v}_{s,s}(\bar{x}; t, x) - \bar{b}_s(\bar{x}; t, x) \right\|^2 ds dt. \tag{22}$$

The expectation is taken over query points $\bar{x}$ and $x$, which can be sampled from data or simulated teacher trajectories.

Minimizing this objective explicitly anchors the MFM's velocity field to the true conditional drift: $\hat{v}_{s,s}(\bar{x}; t, x) = \bar{b}_s(\bar{x}; t, x)$ for all $s, t \in [0, 1]$ and $\bar{x}, x \in \mathbb{R}^d$.

**Consistency Loss.** The availability of $\bar{b}_s$ also stabilizes the consistency objective. We can replace self-consistency targets (specifically terms relying on $\hat{v}_{s,s}$) with fixed, low-variance targets derived directly from $\bar{b}_s$. We detail these specific teacher variants in Table 2.

**MFM Loss.** At every training step, we sample contexts $(t, x)$ and minimize the sum of the diagonal and consistency losses $\mathcal{L}_{\text{MFM}} = \mathcal{L}_{\text{diag}} + \mathcal{L}_{\text{cons}}$. This joint optimization ensures the MFM learns the correct vector field via the diagonal term while enforcing long-range integration accuracy via the consistency term.

### 4.3. Sampling

MFMs enable efficient sampling from the data distribution $p_1$ either through single-step generation or multi-step refinement. For one-shot generation, we draw noise $\epsilon \sim p_0$ and apply the map at time zero, $\hat{x}_1 = X_{0,1}(\epsilon; 0, x)$, which theoretically yields exact samples $\hat{x}_1 \sim p_1$ due to the independence of the prior and data at $t = 0$. To further enhance sample quality, we can employ a $K$-step refinement procedure (Algorithm 6) that iteratively conditions on intermediate states (Appendix E.4).

### 4.4. Inference-Time Steering

We now demonstrate how MFMs enable efficient inference-time steering by estimating the drift $b_t^\star$ for the optimally controlled ODE (10). This requires two components: the base drift $b_t$ and the value function gradient $\nabla_x V_t$.

**Extracting $b_t$.** This drift corresponds to the unconditional flow (3) transporting $p_0$ to $p_1$. Since the conditional distribution $p_{1|0}(\cdot|x_0)$ is simply the marginal $p_1$ for any $x_0$, the flow map $X_{s,u}(\cdot; 0, x_0)$ recovers the unconditional dy-

namics. Using the tangent condition (7), we extract the instantaneous drift for any state $x$ as $b_t(x) = v_{t,t}(x; 0, x_0)$.

**Estimating $\nabla_x V_t$.** The core algorithmic advantage of MFMs lies in the estimation of $\nabla_x V_t$. We implement the estimator $\widehat{\nabla_x V_t(x)}$ in (14) by setting $\Phi = X_{0,1}$, which allows us to draw samples directly from the posterior $X_{0,1}(\epsilon; t, x) \sim p_{1|t}(\cdot|x)$ for $\epsilon \sim p_0$. This approach resolves a major limitation in the steering literature, where prior methods typically lack access to the true conditional posterior and relied on heuristic approximations. For instance, methods such as Bansal et al. (2023); Yu et al. (2023); Chung et al. (2024) approximate the expectation with a point mass at the denoising mean, while Song et al. (2023a) approximates the posterior as a Gaussian centred at the mean. These heuristics introduce significant bias. MFMs overcome this by providing the efficient, differentiable sampling from $p_{1|t}(\cdot|x)$ required to deploy the estimator (14) faithfully.

**Steered ODE.** Substituting $\widehat{\nabla_x V_t}$ into (10) yields a tractable ODE that can be integrated using any standard solver:

$$\dot{x}_t^\star = b_t(x_t^\star) + \frac{\sigma_t^2}{2}\widehat{\nabla V_t(x_t^\star)}, \quad x_0^\star \sim p_0. \quad (23)$$

We detail the procedure in Algorithm 3. While we focus on ODE steering here, MFMs also support SDE steering, see Algorithm 4. We also provide theoretical convergence rates in Appendix H.2, specifically KL and $W_2$ bounds, quantifying errors arising from discretization and Monte Carlo estimation.

**Gradient-Free Steering.** MFMs also support various non-gradient based steering methods. In Appendix C, we present a gradient-free estimator MFM-GF of the optimal drift $b_t^\star$. This estimator can be used in lieu of the MFM-G estimator for steering. However, for improved performance, we employ MFM-Search (Algorithm 1). MFM-Search exploits the efficiency of the MFM to generate a batch of candidate posterior samples at each timestep, selecting the highest-reward candidate to guide the trajectory (see Algorithm 1).

### 4.5. Fine-Tuning

Beyond inference-time steering, MFMs facilitate efficient training-time reward alignment. Given a base drift $b_t$ (potentially extracted from an MFM), we can fine-tune a new model $\hat{b}_t$ to permanently distill the optimal steering drift $b_t^\star$. A naive approach might regress $\hat{b}_t$ directly onto the estimator (14). However, as a *self-normalised* estimator (taking the form of a ratio of sample means), it exhibits a bias for any finite number of Monte Carlo samples $N$. To circumvent this, we construct an unbiased objective. Recall

from (10) and (13) that:

$$b_t^\star(x) = b_t(x) + \frac{\sigma_t^2}{2}\frac{\mathbb{E}_{\epsilon \sim p_0}\left[\nabla_x e^{r(X_{0,1}(\epsilon; t, x))}\right]}{\mathbb{E}_{\epsilon \sim p_0}\left[e^{r(X_{0,1}(\epsilon; t, x))}\right]}. \quad (24)$$

To ensure the learned drift $\hat{b}_t$ matches the optimal $b_t^\star$ without explicitly computing this ratio, we multiply through by the denominator. This yields the following *implicit optimality condition*, which holds if and only if $\hat{b}_t(x) = b_t^\star(x)$:

$$\mathbb{E}_{\epsilon \sim p_0}\left[e^{r(X_{0,1}(\epsilon; t, x))}\left(\hat{b}_t(x) - b_t(x)\right) - \frac{\sigma_t^2}{2}\nabla_x e^{r(X_{0,1}(\epsilon; t, x))}\right] = 0.$$

We enforce this condition with the following objective.

> **Unbiased fine-tuning objective (MFM-FT).** The drift $b_t^\star$ is the unique fixed point of:
>
> $$\mathcal{L}(\hat{b}) := \int_0^1 \mathbb{E}\Big[\big\|(\hat{b}_t(x) - b_t(x)) - \frac{\sigma_t^2}{2}\nabla_x e^{r(X_{0,1}(\epsilon; t, x))}$$
> $$+ (e^{r(X_{0,1}(\epsilon; t, x))} - 1)\,\text{sg}(\hat{b}_t(x) - b_t(x))\big\|^2\Big]dt. \quad (25)$$
>
> The expectation is taken over $\epsilon \sim p_0$ and $x \in \mathbb{R}^d$ can be sampled from any distribution with full support.

Crucially, the MFM-FT objective in (25) is explicitly *off-policy*. The loss is defined pointwise for any $t \in [0, 1]$ and $x \in \mathbb{R}^d$, and so we can sample these from any distribution. In practice, we sample $x$ from the interpolant $I_t$, which is simulation free and should cover the support where we care about learning. This stands in contrast to many fine-tuning methods that rely on on-policy simulation, i.e., they must draw samples from the current model. We can also sample $x$ using the current $\hat{b}_t$ in order to sample $x$ in the most important regions, but this is not necessary.

## 5. Experiments

We evaluate the capabilities of Meta Flow Maps (MFMs) along three axes: the fidelity of posterior sampling and the performance of inference-time steering and fine-tuning. We begin with inference-time steering experiments where the reward function is defined through a likelihood, targeting posterior distributions. We then consider ImageNet ($256 \times 256$), where we demonstrate the efficacy of both inference-time steering and fine-tuning at scale using text-to-image reward functions. Additional experimental details and results are found in Appendix K.

### 5.1. Gaussian Mixture Models (GMMs)

We first consider a linear inverse problem, where the goal is to sample the posterior $p(x|y_{\text{obs}})$ over a latent variable $x$ given a noisy, transformed observation $y_{\text{obs}}$. We use a three-component two-dimensional Gaussian mixture model (GMM) as the prior $p(x)$, together with a linear Gaussian

observation model $p(y_{\text{obs}}|x) = \mathcal{N}(y_{\text{obs}}; Ax, \sigma^2 I)$, where $A$ denotes the linear measurement operator. We train a MFM to sample the GMM prior $p(x)$, and then use the log-likelihood as our reward for inference-time steering. Crucially, the posterior $p(x|y_{obs})$ is available in closed form, allowing us to quantify the distributional error of samples from various steering algorithms.

We compare our gradient-based (MFM-G;(14)) and gradient-free (MFM-GF; (32)) estimators against Diffusion Posterior Sampling (DPS; (Chung et al., 2024)) and Sequential Monte Carlo (SMC-TDS; Wu et al. (2024)). DPS fails to capture the full posterior variance, significantly over-representing the dominant mode (Figure 11). By contrast, MFM-based steering accurately recovers the true posterior distribution, outperforming DPS and SMC-TDS even with as few as $N = 2$ MC samples (Figure 5). Empirically, we also observe strong scaling with the number of MC samples for the MFM samplers. We observe improvement as measured by maximum mean discrepancy (MMD) and the Sliced-Wasserstein distances to the ground truth (Figure 10).

## 5.2. MNIST

Next, we consider a conditional sampling task on MNIST designed to investigate performance in a highly multimodal setting. We define the reward function as a weighted sum of class probabilities as predicted by a classifier (Appendix K.2). Although the MNIST dataset exhibits approximately uniform class frequencies, this reward induces a target distribution with strongly imbalanced class proportions. The resulting challenge is to generate samples that accurately match the specified target ratios across digits.

Consistent with the GMM results, DPS fails to respect the target ratios, collapsing towards the dominant mode (Figures 5, 12). By contrast, both MFM-G and MFM-GF approach the correct target distribution as the number of MC samples increases. Notably, in this higher-dimensional setting, the gradient-based estimator (MFM-G) significantly outperforms the gradient-free variant (MFM-GF).

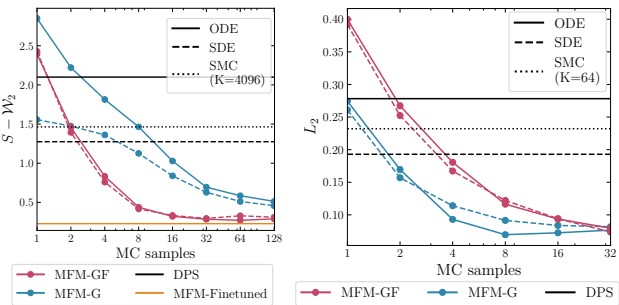

*Figure 5.* **Left:** Comparison for GMM inverse problem: Sliced-Wasserstein, $\mathcal{S}\text{-}\mathcal{W}_2$, between true posterior samples and steered samples. **Right:** Comparison for MNIST sampling problem: $L_2$ between the target and empirical class ratios of steered samples.

## 5.3. ImageNet

**Model.** We scale MFMs to ImageNet ($256 \times 256$) using a standard latent Diffusion Transformer (DiT; Peebles & Xie (2023)). We minimally modify the architecture to accept the time-state $(t, x)$ conditioning required for MFMs; this overhead results in only a 1.3% increase in parameters (Appendix G.2). As our primary goal is to demonstrate the efficacy of inference-time steering and fine-tuning, we defer a detailed discussion of training specifications to Appendix K, where we present results for MFMs trained through both data and distillation. In what follows, we present the results of our most performant checkpoint obtained through fine-tuning[1] DMF-XL/2+ (Lee et al., 2025), a state-of-the-art deterministic flow map. We train it using the Eulerian-Teacher objective in Table 2.

**Unconditional Generation.** Table 1 presents the performance of MFM-XL/2 using the $K$-step refinement sampler in Algorithm 6. We achieve a competitive FID of **1.97** in just 4 steps, showing that MFMs are competitive with state-of-the-art deterministic baselines, while additionally providing stochastic one-step posterior samples for reward alignment.

*Table 1.* ImageNet ($256 \times 256$) benchmark.

| **Deterministic Few-Step Flow Models** | | | |
|---|---|---|---|
| Method | NFE | #Params | FID $\downarrow$ |
| Shortcut-XL/2 | 1 | 676M | 10.60 |
| (Frans et al., 2025) | 4 | 676M | 7.80 |
| IMM-XL/2 | $2 \times 1$ | 676M | 7.77 |
| (Zhou et al., 2025) | $2 \times 2$ | 676M | 3.99 |
| | $2 \times 4$ | 676M | 2.51 |
| MF-XL/2+ | 1 | 676M | 3.43 |
| (Geng et al., 2025a) | 2 | 676M | 2.20 |
| DMF-XL/2+ | 1 | 675M | 2.16 |
| (Lee et al., 2025) | 2 | 675M | 1.64 |
| | 4 | 675M | 1.51 |
| **Stochastic Few-Step Flow Models** | | | |
| **MFM-XL/2** | 1 | 683M | 3.72 |
| | 2 | 683M | 2.40 |
| | 4 | 683M | 1.97 |

**Posterior Fidelity.** Next, we evaluate posterior samples $\hat{x}_1 \sim p_{1|t}(\cdot|x_t)$ from the MFM against those obtained through ODE rollouts of the conditional drift extracted from DMF-XL/2+ using GLASS flows. Note that this drift is the source of diagonal supervision for MFM-XL/2, and as such,

---

[1]For an overview of the computational costs of training, the base flow matching (SIT) takes 800 epochs (Ma et al., 2024), the DMF flow map training takes an additional 80 epochs (Lee et al., 2025), and the MFM training takes an additional 30 epochs.

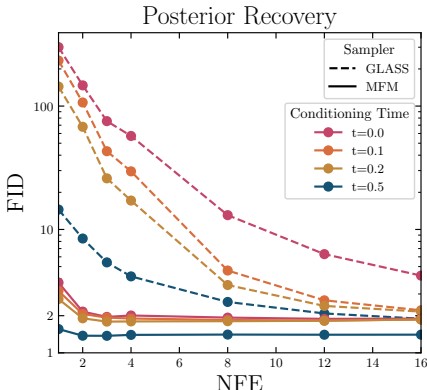
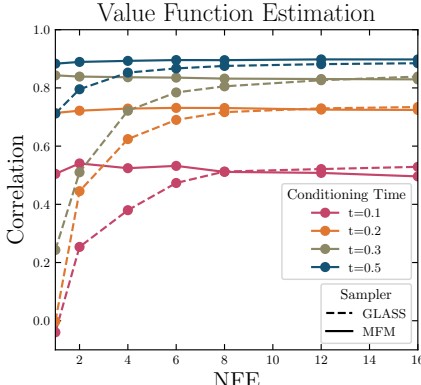

*Figure 6.* Posterior sampling performance of MFM-XL/2 and GLASS flows for different conditioning times, $t$, and NFE budgets. **Left** graph shows the posterior FID. **Right** graph shows the correlation (Pearson's $r$) between an expensive, high-fidelity MC estimator (SDE rollout) and an estimator obtained through MFM or GLASS flows.

allows for a direct evaluation of the computational benefits of learning a one-step sampler. Across conditioning times ($t$) and number of function evaluations (NFEs), we evaluate performance using the following two metrics:

**(A) Posterior Recovery.** Starting from clean ImageNet images $\{x_1^{(i)}\}$, we noise to time $t$ to obtain $\{x_t^{(i)}\}$. For each noisy input, we draw a single posterior sample $\hat{x}_1^{(i)} \sim p_{1|t}(\cdot|x_t^{(i)})$ using either the MFM or ODE rollouts using GLASS flows. We then compute the FID between the resulting set $\{\hat{x}_1^{(i)}\}$ and the original ImageNet images $\{x_1^{(i)}\}$. This evaluates how faithfully each conditional sampler recovers the data distribution from intermediate noise levels.

**(B) Value Function Estimation.** We also evaluate posterior samplers by their ability to support Monte Carlo estimation of expectations under $p_{1|t}(\cdot|x)$, focusing on the value function $V_t(x) = \log \mathbb{E}_{x_1 \sim p_{1|t}(\cdot|x)}[\exp(r(x_1))]$. We form Monte Carlo estimates of $V_t(x)$ using posterior samples generated by either MFM or GLASS ODE rollouts. Estimation accuracy is quantified via correlation with a high-fidelity reference estimator obtained from an expensive SDE rollout which we treat as the ground truth. This isolates how well each sampler computes posterior expectations.

MFMs strongly outperform explicit ODE rollouts on both **(A)** and **(B)**, across conditioning times and NFEs (Figure 6). Notably, the improvement is greatest at a 1-NFE budget, which is of particular interest, as differentiating through rollouts to realise the tilted drift estimator in (14) is prohibitively expensive. See Appendix K.3.2 for further details on these quantitative metrics.

### 5.3.1. INFERENCE-TIME STEERING

Having demonstrated the substantial computational advantages of training MFMs for few-step posterior sampling, we

now return to the core motivation: reward alignment. We begin by evaluating inference-time steering, where we steer the MFM-XL/2 model using text-to-image reward models.

**Reward.** We steer the class-conditioned ODE targeting class *tabby cat* using ImageReward (Xu et al., 2023), Pick Score (Kirstain et al., 2023), and HPSv2 (Wu et al., 2023) conditioned on the prompt *"A high-quality, high-resolution photograph of a tabby cat."*. We consider a range of reward multipliers, $\lambda = \{1.0, 2.5, 5.0\}$, where $p_{\text{reward}}(x) \propto p_{\text{model}}(x) \exp(\lambda r_\theta(x, \text{prompt}))$. Further experimental details can be found in Appendix K.

**Methods.** We present MFM-G and MFM-Search (Algorithm 1). For clarity of presentation, we omit MFM-GF from the main figures due to consistently inferior performance over the alternative gradient-free solution, MFM-Search; complete results are provided in Appendix K.3.3. We compare against DPS and the Best-of-N (BoN) baseline, which generates $N_{\text{BoN}}$ samples and selects the highest-reward sample. For each method, we generate 128 images and report the average reward.

**Compute-Normalised Performance.** We consider performance as a function of the number of function evaluations (NFEs) (see Appendix K.3.6 for a detailed count of the NFEs). MFM-G achieves substantially better compute-normalised performance than Best-of-$N$ and DPS across all reward models (Figure 7). In particular, its performance curves lie strictly above those of Best-of-$N$. Notably, even the cheapest MFM-G variant ($N = 1$) outperforms the most expensive Best-of-$N$ configuration ($N_{\text{BoN}} = 1000$), while requiring over $100\times$ fewer NFEs. To validate that the observed improvements are not driven by reward hacking, we evaluate performance using reward models different from the one used for steering (Appendix K.3.3). We find that steering with any of the reward models does not degrade performance on the remaining metrics, and in many

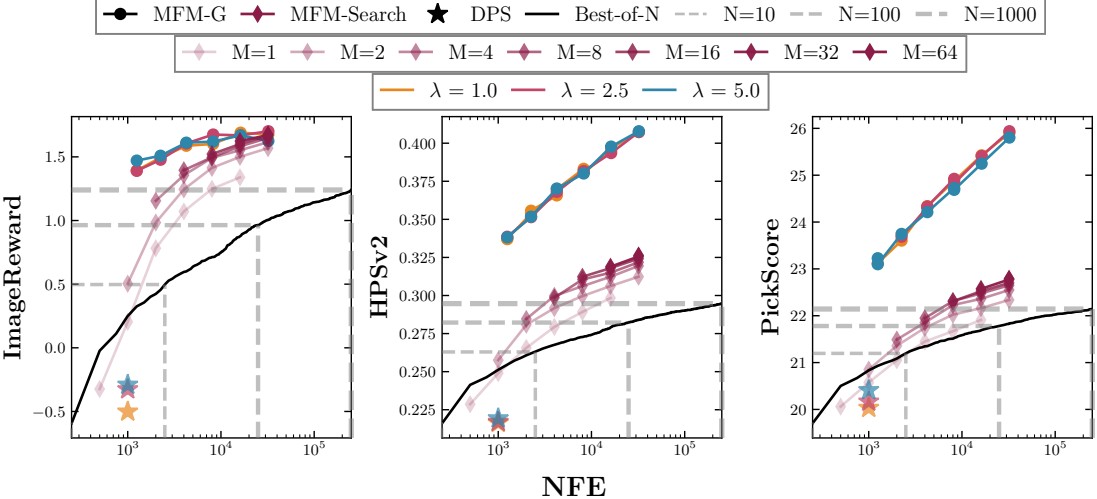

*Figure 7.* Compute-normalised performance comparison of inference-time steering schemes.

cases yields substantial improvements. In particular, when steering with either HPSv2 or PickScore, the resulting generations achieve higher scores on the other reward model than $N_{\text{BoN}} = 1000$.

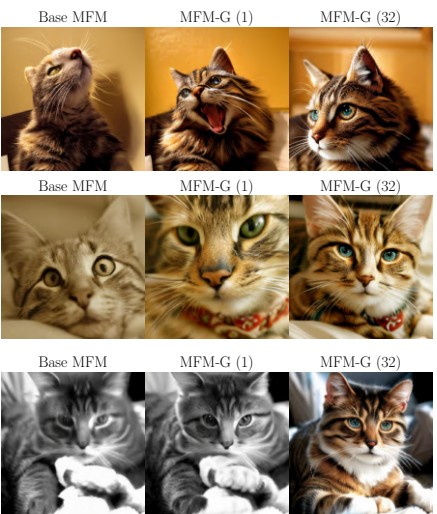

*Figure 8.* Base and steered (HPSv2) samples. A larger set of samples is provided in Figure 20.

### 5.3.2. FINE-TUNING

Finally, we consider fine-tuning using the objective in (25). To this end, we fine-tune across all the ImageNet classes using the HPSv2 reward model, conditioned on the prompt template *"A high-quality, high-resolution photograph of a {class}."*, aiming to generally enhance the aesthetics of samples. We evaluate performance across a range of reward multipliers, $\lambda = \{10, 25, 50\}$ in Figure 9.

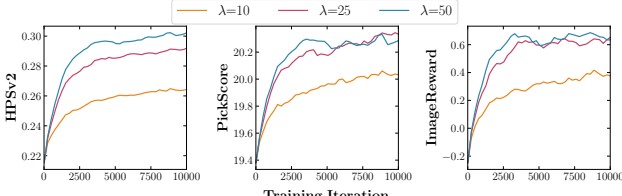

*Figure 9.* Rewards on HPSv2, ImageReward, and PickScore during fine-tuning with the HPSv2 reward model. Metrics are evaluated every 500 training iterations and averaged over 512 ODE samples.

Figure 9 demonstrates the effectiveness of the fine-tuning procedure. We emphasize that we train *only* on HPSv2, but we report scores on all three reward models to ensure that we are not reward hacking. These improvements are also qualitatively reflected in the samples shown in Figure 3, where the fine-tuned MFMs produce visibly higher-quality and more colourful images while preserving the semantic content of the base MFM samples.

## 6. Conclusion

In this work, we introduced Meta Flow Maps, a stochastic extension of consistency models and flow maps that makes diffusion models controllable by design through consistent inference-time steering and off-policy fine-tuning. While our experiments focus on images, the framework opens the door to scalable plug-and-play reward alignment across continuous diffusion and flow-based generative models, including text, video, robotics, and molecular generation. In these domains, the ability to redirect pre-trained generators toward new rewards, constraints, and user preferences after training is of growing importance as flow models become increasingly prevalent in downstream applications.

## Impact Statement

This paper presents work whose goal is to advance the field of Machine Learning. There are many potential societal consequences of our work, none which we feel must be specifically highlighted here.

## Acknowledgments

We thank Iskander Azangulov, Jakiw Pidstrigach, Sam Howard, Franklin Shiyi Wang, Yuyuan Chen, Carles Domingo-Enrich, Francisco Vargas, Peter Holderrieth, and George Deligiannidis for fruitful conversations. We also want to thank Arnaud Doucet in particular for his help with the theoretical development and for his suggestions on the paper. PP is supported by the EPSRC CDT in Modern Statistics and Statistical Machine Learning [EP/S023151/1], a Google PhD Fellowship, and an NSERC Postgraduate Scholarship (PGS D). AS is supported by the EPSRC CDT in Modern Statistics and Statistical Machine Learning [EP/Y034813/1]. AM is supported by the Clarendon Fund Scholarship, University of Oxford. AP is supported by the EPSRC and AstraZeneca via an iCASE award for a DPhil in Machine Learning. MSA is supported by a Junior Fellowship at the Harvard Society of Fellows as well as the National Science Foundation under Cooperative Agreement PHY-2019786 (The NSF AI Institute for Artificial Intelligence and Fundamental Interactions[2]). This work has been made possible in part by a gift from the Chan Zuckerberg Initiative Foundation to establish the Kempner Institute for the Study of Natural and Artificial Intelligence. The authors also acknowledge the use of resources provided by the Isambard-AI National AI Research Resource (AIRR) (McIntosh-Smith et al., 2024). Isambard-AI is operated by the University of Bristol and is funded by the UK Government's Department for Science, Innovation and Technology (DSIT) via UK Research and Innovation; and the Science and Technology Facilities Council [ST/AIRR/I-A-I/1023].

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

# A. Related Work

**Inference-Time Alignment & Steering.**  Inference-time methods aim to adapt the sampling dynamics of a fixed pretrained model to sample from the reward-tilted distribution $p_{\text{reward}}$. Existing approaches broadly fall into two categories: methods that aim to approximate the exact tilted dynamics, and particle-based methods that rely on resampling or search.  For the first class, many methods can be seen as attempting to perform exact steering by approximating the true posterior distribution $p_{1|t}(\cdot|x)$ with a surrogate. One common heuristic is to approximate this posterior with a point mass (such as at the conditional expectation of data, i.e., the denoised estimate) as done in DPS (Chung et al., 2024), FreeDoM (Yu et al., 2023), MPGD (He et al., 2023), or unified frameworks based on a similar principle (Bansal et al., 2023; Ye et al., 2024). Other methods attempt to have a more refined approximation to this posterior, such as LGD (Song et al., 2023a), which approximates the posterior as a Gaussian centred at the posterior mean with a manually selected variance. While efficient, these approximations introduce significant bias and often fail to guide trajectories correctly in multimodal or non-linear settings where the mean does not represent a valid data sample (He et al., 2023).

Conversely, exact sampling methods such as Sequential Monte Carlo (SMC) reweight or resample trajectories to strictly target the tilted distribution (Wu et al., 2024; Skreta et al., 2025; Singhal et al., 2025). Although unbiased in principle, these methods require a prohibitively large number of particles to avoid weight degeneracy and collapse (Snyder et al., 2008; Bickel et al., 2008). Recent search-based methods attempt to mitigate this by estimating intermediate rewards via explicit rollouts (Li et al., 2025; Zhang et al., 2025), but this still incurs a substantial computational cost per sampling step. We refer to (Uehara et al., 2025) for an overview.

**Few-Step Samplers.**  Our training scheme for MFMs leverages learning objectives from the literature on consistency models (Song et al., 2023b; Song & Dhariwal, 2023) and flow maps (Kim et al., 2024a; Frans et al., 2024; Geng et al., 2024; 2025b; Boffi et al., 2025b;a; Sabour et al., 2025).  However, unlike existing approaches, which aim to accelerate sampling, MFMs must provide access to cheap (one-step), differentiable samples from $p_{1|t}(\cdot|x)$, for all $(t, x)$ to support efficient estimation of the tilted drift.

**Posterior Sampling.**  A critical bottleneck in exact steering is the need to efficiently sample from the conditional posterior $p_{1|t}(\cdot|x)$. Many prior works rely on trajectory rollouts of SDEs (Elata et al., 2023; Li et al., 2025; Zhang et al., 2025; Jain et al., 2025). Due to the inefficiencies of SDE sampling, recent work has introduced training-free methods to enable more efficient ODE samplers. In the case where the prior is Gaussian, GLASS Flows (Holderrieth et al., 2025) leverage the sufficient statistics of Gaussian integrals to reparameterize standard pretrained models into transition samplers. However, this method still relies on solving expensive ODEs during inference. Unlike GLASS, MFMs eliminate the need for this iterative integration. Alternative approaches explicitly learn these transitions during training. For instance, Distributional Diffusion (Bortoli et al., 2025) and Gaussian Mixture Flow Matching (Chen et al., 2025) train models to output posterior distributions directly via proper scoring rules or mixture approximations. While Distributional Diffusion also trains a stochastic flow map, training with scoring rules can be challenging to tune and scale. In particular, they were not successful in implementing their approach for ImageNet ($64 \times 64$), while we show MFMs scale successfully to ImageNet ($256 \times 256$).

**Value Function Estimation.**  We mention the closely related field of neural sampling. Neural samplers aim to generate samples from a target distribution $p_1$ given access only to its unnormalised density $\tilde{p}_1$. These methods are related to steering algorithms as sampling can often be rephrased as modifying a process that targets a simple reference distribution $p_{\text{ref}}$ and steering it to sample from $p_1$ by choosing the reward $r(x) = \log \tilde{p}_1(x) - \log p_{\text{ref}}(x)$. In this way, steering algorithms and neural samplers can often be repurposed for the reciprocal task. Many existing neural samplers approach sampling as a stochastic optimal control problem and therefore also require obtaining estimates of the value function and its gradient. Within this field, we highlight a class of methods that use gradient-free Monte Carlo estimators for these objects (Huang et al., 2021; Vargas et al., 2022; Akhound-Sadegh et al., 2024). However, these methods do not have access to true data samples and so they cannot obtain posterior samples directly. As a result, they often suffer from high variance.

**Generative Fine-Tuning.**  Beyond inference-time steering, permanent weight adaptation is another dominant strategy for alignment. Existing work broadly follows two paradigms: *(A) Reward maximization* such as D-Flow (Ben-Hamu et al., 2024) and DRaFT (Clark et al., 2024), that directly optimize the expected reward. However, this often leads to mode collapse and over-fitting (Goodhart's Law) as the model collapses to a single high-reward mode rather than the true posterior; *(B) Distribution Matching* techniques aim to align the model with the reward-tilted distribution, preserving diversity. Notable examples include DEFT (Denker et al., 2025), Adjoint Matching (Domingo-Enrich et al., 2025), Tilt Matching (Potaptchik

et al., 2025) and diffusion variants of DPO (Wallace et al., 2023).

## B. Controlling Dynamics via Doob's $h$-Transform

We present below further details on steering the unconditional dynamics of SDEs and ODEs to sample the target distribution, $p_{\text{reward}}(x) \propto p_{\text{model}}(x)e^{r(x)}$.

### B.1. SDE Steering

Recall that the drift $b_t$ in (3) was chosen such that the ODE marginals match the interpolant's: $\text{Law}(x_t) = \text{Law}(I_t) = p_t$. A standard approach to obtain stochastic dynamics with these same marginals is to introduce diffusion while compensating in the drift (Song et al., 2020; Albergo et al., 2023a), yielding the following SDE:

$$dX_t = \left[b_t(X_t) + \tfrac{\sigma_t^2}{2}\nabla \log p_t(X_t)\right] dt + \sigma_t\, dB_t, \qquad X_0 \sim p_0, \tag{26}$$

where the diffusion coefficient is $\frac{\sigma_t^2}{2} = \frac{\dot{\beta}_t}{\beta_t}\alpha_t^2 - \dot{\alpha}_t\alpha_t$. While any diffusion schedule $\sigma_t$ in this formulation yields the correct marginals $\text{Law}(X_t) = p_t$, the SDE $X_t$ and the interpolant $I_t$ generally possess different transition kernels. We employ this specific schedule because it ensures that the conditional endpoints laws are identical; that is, the distribution of $X_1$ conditioned on $X_t = x$ matches that of $I_1$ conditioned on $I_t = x$ (we verify this in Appendix H). Consequently, we let $p_{1|t}(\cdot|x)$ denote the density of this shared *conditional posterior distribution*:

$$p_{1|t}(\cdot|x) = \text{Law}(X_1 \,|\, X_t = x) = \text{Law}(I_1 \,|\, I_t = x). \tag{27}$$

For the SDE (26), we recall the *value function* $V_t(x)$ defined in (2):

$$V_t(x) = \log \mathbb{E}\big[e^{r(X_1)}|X_t = x\big] = \log \mathbb{E}\big[e^{r(I_1)}|I_t = x\big], \tag{28}$$

where the second equality holds precisely because the SDE and the interpolant share identical conditional endpoint laws. Doob's $h$-transform (Dai Pra, 1991; Denker et al., 2025) tilts the path measure by the terminal reward $e^{r(X_1)}$ by adding the diffusion-scaled gradient of the value function to the drift, yielding the optimally controlled SDE:

$$dX_t^\star = \left[b_t(X_t^\star) + \tfrac{\sigma_t^2}{2}\nabla \log p_t(X_t^\star) + \sigma_t^2 \nabla V_t(X_t^\star)\right] dt + \sigma_t\, dB_t, \qquad X_0^\star \sim p_0. \tag{29}$$

Crucially, the score term in (29) corresponds to the uncontrolled process and is therefore already available; when $p_0$ is Gaussian, the score $\nabla \log p_t$ is simply a linear reparameterization of $b_t$. This leaves $\nabla V_t$ as the only missing component. Under optimal control, the marginal density $p_t^\star$ of $X_t^\star$ satisfies

$$p_t^\star(x) \propto p_t(x)e^{V_t(x)}, \tag{30}$$

ensuring the terminal marginal is exactly $p_1^\star = p_{\text{reward}}$.

### B.2. ODE Steering

We can also define the corresponding probability flow ODE to (29), whose trajectories satisfy $\text{Law}(x_t^\star) = \text{Law}(X_t^\star) = p_t^\star$, by subtracting half the diffusion-scaled score from the drift:

$$\dot{x}_t^\star = \underbrace{b_t(x_t^\star) + \tfrac{\sigma_t^2}{2}\nabla V_t(x_t^\star)}_{b_t^\star(x_t^\star)}, \quad x_0^\star \sim p_0. \tag{31}$$

The optimal drift $b_t^\star$ serves as the target for both control paradigms: it can be estimated to steer trajectories during inference, or distilled permanently into a student model during fine-tuning. Since $b_t$ is readily available from the pretrained model, the central algorithmic challenge is efficiently estimating the value function gradient $\nabla V_t(x)$.

## C. Gradient-Free Estimator

We present below a estimator for $\nabla V_t(x)$, adapted from Potaptchik et al. (2025), that requires only reward function evaluations and i.i.d. samples from the posterior $p_{1|t}(\cdot|x)$.

**Gradient-Free Estimator (MFM-GF).** A consistent Monte Carlo estimator of $\nabla V_t(x)$ is

$$\frac{\sigma_t^2}{2} \widehat{\nabla V_t(x)} = \left( \dot{\beta}_t - \frac{\dot{\alpha}_t}{\alpha_t} \beta_t \right) \frac{\sum_{i=1}^N x_1^{(i)} \exp(r(x_1^{(i)}))}{\sum_{i=1}^N \exp(r(x_1^{(i)}))} + \frac{\dot{\alpha}_t}{\alpha_t} x - b_t(x), \qquad x_1^{(i)} \stackrel{\text{iid}}{\sim} p_{1|t}(\cdot|x). \tag{32}$$

Unlike the efficient gradient estimator (14) presented in the main body, this estimator is compatible with non-differentiable reward functions.

## D. Limitations of Existing Posterior Samplers

As covered in the main body, obtaining posterior samples from $p_{1|t}(\cdot|x)$ to Monte Carlo estimate the gradient of the value function $\nabla_x V_t$ is a major bottleneck in both inference-time steering, and fine-tuning. Existing approaches typically rely on expensive trajectory unrolling, while standard acceleration techniques like consistency models and flow maps are structurally ill-suited for conditional sampling.

**Inner rollouts of SDEs.** A direct way to obtain posterior samples is via "inner rollouts", where one simulates the SDE (26) forward from time $t$ to 1, starting at $x$ (Elata et al., 2023; Li et al., 2025; Zhang et al., 2025; Jain et al., 2025). By repeating this process with independent noise, one yields a batch of samples from $p_{1|t}(\cdot|x)$. However, this approach is prohibitively costly, as it nests a full inner simulation within every step of the outer steering trajectory or every step of fine-tuning.

**Inner rollouts of ODEs.** In principle, SDE simulation can be replaced with ODE sampling. As noted in the main body, although the conditional drift targeting the posterior $p_{1|t}(.|x)$ is well defined in theory, it is generally intractable without retraining. However, when the prior $p_0$ is Gaussian, GLASS flows (Holderrieth et al., 2025) demonstrate that $\bar{b}_s$ can be derived analytically by reparameterizing the drift $b_t$ (21) (See Appendix I for the explicit form of the coefficients). While this reparametrization makes $\bar{b}_s$ accessible, generating posterior samples from $p_{1|t}(\cdot|x)$ by unrolling ODE trajectories still remains computationally expensive. It requires drawing a separate initial condition $\bar{x}_0 \sim p_0$ for every element in the Monte Carlo batch and integrating (16).

Furthermore, efficient estimators of $\nabla V_t(x)$, such as (14), also require that the posterior samples $x_1 \sim p_{1|t}(\cdot|x)$ are differentiable with respect to $x$ via the reparametrization $\Phi$. While it is theoretically possible to differentiate through the ODE or SDE solvers used in exact rollouts, this incurs a prohibitive memory and computational cost, making it impractical for iterative steering or fine-tuning. Consequently, this has often forced reliance on coarse approximations.

**Insufficiency of flow maps.** Finally, we remark that standard flow maps address a fundamentally different transport problem than what is required for posterior sampling. A flow map $X_{s,u}$ is trained to satisfy the *marginal* transport constraint:

$$X_{s,u} \# p_s = p_u, \qquad \forall s, u \in [0,1]. \tag{33}$$

Because the map $x \mapsto X_{t,1}(x)$ is a deterministic function of $x$, it is structurally incapable of representing the full conditional posterior $p_{1|t}(\cdot|x)$, which generally admits a distribution of valid endpoints for any fixed intermediate state $x$ at time $t$.

This leaves us with a fundamental dilemma. Exact sampling methods, such as SDE or ODE rollouts, can capture the diversity of the posterior but are prohibitively slow due to the need for iterative integration. Conversely, accelerated methods like consistency models and flow maps are efficient but, due to their deterministic nature, cannot capture the stochasticity required for posterior sampling.

## E. Methodology

### E.1. Reparametrization

Suppose that we have an MFM $X$ trained on an interpolant $(I_t)_{t \in [0,1]}$ with coefficients $\alpha_t, \beta_t$. We emphasize that here we do not necessarily assume that $p_0$ is Gaussian. We denote the marginal density of $I_t$ by $p_t$ and the conditionals posterior of $I_1$ given $I_t = x$ by $p_{1|t}(\cdot|x)$. We describe how this MFM can be be reparametrized to sample from posteriors arising from interpolants with different coefficients. Let $\tilde{I}_t$ be the interpolant defined by

$$\tilde{I}_t = \tilde{\alpha}_t I_0 + \tilde{\beta}_t I_1, \tag{34}$$

for some new coefficients $\tilde{\alpha}_t, \tilde{\beta}_t$. Let $\tilde{p}_t$ be the marginal density of $\tilde{I}_t$ and let $\tilde{p}_{1|t}(\cdot|x)$ denote the conditional posterior. Rearranging the interpolant definition, we have

$$\frac{1}{\tilde{\beta}_t}\tilde{I}_t = \frac{\tilde{\alpha}_t}{\tilde{\beta}_t}I_0 + I_1. \tag{35}$$

Since $\alpha_t$ and $\beta_t$ are continuous in $t$ and since the boundary conditions satisfy $\alpha_0 = 1, \beta_0 = 0$ and $\alpha_1 = 0, \beta_1 = 1$, this implies that the map $t \mapsto \frac{\alpha_t}{\beta_t}$ is a surjection from $[0, 1]$ onto $[0, \infty)$. Therefore there exists $t^* \in [0, 1]$ such that $\frac{\alpha_{t^*}}{\beta_{t^*}} = \frac{\tilde{\alpha}_t}{\tilde{\beta}_t}$ and for this $t^*$ we have

$$\frac{1}{\tilde{\beta}_t}\tilde{I}_t = \frac{\alpha_{t^*}}{\beta_{t^*}}I_0 + I_1, \tag{36}$$

and so

$$\frac{\beta_{t^*}}{\tilde{\beta}_t}\tilde{I}_t = \alpha_{t^*}I_0 + \beta_{t^*}I_1. \tag{37}$$

In particular, this shows that

$$\tilde{p}_{1|t}(\cdot|x) = p_{1|t^*}(\cdot|\frac{\beta_{t^*}}{\tilde{\beta}_t}x). \tag{38}$$

Therefore, we can sample from the conditional posterior $\tilde{p}_{1|t}(\cdot|x)$ using the MFM $X$ trained on the interpolant with coefficients $\alpha_t, \beta_t$. In particular, if $\epsilon \sim p_0$, then

$$X_{0,1}(\epsilon; t^*, \frac{\beta_{t^*}}{\tilde{\beta}_t}x) \sim \tilde{p}_{1|t}(\cdot|x). \tag{39}$$

This means that we can obtain differentiable, one-shot samples from posteriors of this form even if our MFM was trained on a different interpolant path.

### E.2. Short Flow Segments

In practice, we will often explicitly form $b_t(x)$ when performing Euler-style updates for an ODE or SDE simulation. As discussed in Section 4.4, we can extract the unconditional drift from a trained MFM.

Another approach is to use short flow segments as follows:

$$\Delta t\, b_t(x) = X_{t,t+\Delta t}(x; 0, x_0) - x + \mathcal{O}(\Delta t^2), \tag{40}$$

which holds for any $x, x_0 \in \mathbb{R}^d$. Depending on the context, this may help reduce discretization errors.

### E.3. Drift Reparametrization

In addition to the extraction in Section (4.4), another reparametrization that can be used to extract the unconditional drift $b_t$ from an MFM $X$ is given by

$$b_t(x_t) = \frac{\dot{\alpha}_t}{\alpha_t}x_t + \frac{\dot{\beta}_t - \frac{\dot{\alpha}_t}{\alpha_t}\beta_t}{\dot{\beta}_0}\left(v_{0,0}(x; t, x_t) - \dot{\alpha}_0 x\right), \tag{41}$$

which holds for any $x$ and any $x_t$. This follows from the identities $b_t(x_t) = \frac{\dot{\alpha}_t}{\alpha_t}x_t + (\dot{\beta}_t - \frac{\dot{\alpha}_t}{\alpha_t}\beta_t)\mathbb{E}[I_1|I_t = x_t]$ and $v_{0,0}(x, t, x_t) = \dot{\alpha}_0 x + \dot{\beta}_0\mathbb{E}[I_1|I_t = x_t]$.

### E.4. $K-$step sampler.

We can sample using a $K$-step refinement procedure as outlined in Algorithm 6. With a well-trained MFM, each $\hat{x}_1^{(k)} \sim p_1$, so $x_{t_{k+1}}$ has marginal density $p_{t_{k+1}}$ for all $k$. This iterative procedure improves sample quality, as it increasingly relies on one-step maps at larger conditioning times $t$, which are typically easier to learn accurately. While we show the algorithm using the same Gaussian path as the original interpolant (4), by reparametrization the same $K$-step sampler applies to any other interpolant path $\tilde{\alpha}_t, \tilde{\beta}_t$ (see Appendix E.1). See Appendix E.5 for a discussion on the connection to $\gamma$-sampling.

**E.5. Connection to $\gamma$-sampling.**

We note that our $K$-step refinement sampler is similar to flow map $\gamma$-sampling (Kim et al., 2024b) with $\gamma = 1$. See Algorithm 7 for an implementation with MFMs. In particular, the difference is that the step $\hat{x}_1^{(k)} \leftarrow X_{0,1}(\epsilon^{(k)}; t_k, x_{t_k})$ is replaced with

$$\hat{x}_1^{(k)} \leftarrow X_{t_k,1}(x_{t_k}; 0, \vec{0}). \tag{42}$$

As with $K$-step refinement, we have that $\hat{x}_1^{(k)} \sim p_1$, so $x_{t_{k+1}}$ has marginal density $p_{t_{k+1}}$ for all $k$. However, the core difference is that $\gamma$-sampling employs marginal transport whereas the $K$-step refinement uses conditional transport to obtain the sample at time 1.

**E.6. Extension to Arbitrary Prediction Times and General Stochastic Processes**

The MFM construction generalizes beyond fixed-endpoint generation to support prediction at arbitrary intermediate times for stochastic processes defined over a general index set $\mathcal{T}$. One can learn MFMs to condition on multiple time points $\{t_i\}_{i=1}^M$, for simplicity, we detail the case of conditioning on one intermediate time $t$. Let $(X_t)_{t \in \mathcal{T}}$ be an arbitrary stochastic process; for example, $(X_t)$ could represent frames in a video or a time series of weather data. We emphasize that this process need not be an interpolant or defined by a flow matching process. Fix a conditioning pair $(t, x) \in \mathcal{T} \times \mathbb{R}^d$ and a target prediction time $r \in \mathcal{T}$. We define the conditional distribution as

$$p_{r|t}(\cdot|x) := \text{Law}(X_r|X_t = x). \tag{43}$$

As in Section 4, we introduce a context-dependent drift $\bar{b}_s(\cdot; r, t, x)$ defined over an auxiliary flow time $s \in [0, 1]$. This drift is defined as the solution to a conditional flow matching problem that transports a simple base noise distribution $q$ to the target posterior $p_{r|t}(\cdot|x)$. We choose the initial distribution of this auxiliary flow to be a tractable distribution $q$, such as a Gaussian. (Note that 0 may not be in the index set $\mathcal{T}$ and even if it is, $p_0 = \text{Law}(X_0)$ is generally intractable to sample from during inference. This is why we choose a different base measure $q$.) This drift defines an auxiliary probability flow ODE

$$\frac{d}{ds}\bar{x}_s = \bar{b}_s(\bar{x}_s; r, t, x), \quad \bar{x}_0 \sim q \quad \implies \quad \text{Law}(\bar{x}_1) = p_{r|t}(\cdot|x). \tag{44}$$

We emphasize that the auxiliary flow $(\bar{x}_s)_{s \in [0,1]}$ does not reproduce the conditional physical evolution of the process $(X_\tau)$ from $\tau = t$ to $\tau = r$; instead, it serves strictly as a transport bridge constructed to satisfy the endpoint distributional constraint $\bar{x}_1 \sim p_{r|t}(\cdot|x)$. We define an *extended Meta Flow Map* $X_{s,u}(\cdot; r, t, x) : \mathbb{R}^d \to \mathbb{R}^d$ as the parametric solution operator for this infinite family of ODEs, satisfying

$$X_{s,u}(\bar{x}_s; r, t, x) = \bar{x}_u, \qquad \forall s, u \in [0, 1], \tag{45}$$

where $(x_\tau)_{\tau \in [0,1]}$ are trajectories of (44). Consequently, $X$ satisfies the property of an extended stochastic flow map

$$X_{0,1}(\cdot; r, t, x) \# q = p_{r|t}(\cdot|x), \qquad \forall r, t \in \mathcal{T}, \forall x \in \mathbb{R}^d. \tag{46}$$

In practice, we parametrize the MFM in terms of the average velocity $v_{s,u}(\cdot; r, t, x) : \mathbb{R}^d \to \mathbb{R}^d$:

$$\hat{X}_{s,u}(\bar{x}; r, t, x) = \bar{x} + (u - s)v_{s,u}(\bar{x}; r, t, x). \tag{47}$$

Training requires minimizing a diagonal loss $\mathcal{L}_{\text{diag}}$ and a consistency loss $\mathcal{L}_{\text{cons}}$ over a neural parameterization $\hat{v}_{s,u}$. To train the diagonal, we sample coupled pairs $(X_r, X_t)$ from a dataset of real trajectories and construct a reference interpolant

$$\bar{I}_s = \alpha_s \bar{I}_0 + \beta_s Y_r, \qquad \bar{I}_0 \sim q. \tag{48}$$

The flow matching loss is evaluated by regressing the learned velocity onto the time derivative $\frac{d}{ds}\bar{I}_s = \dot{\alpha}_s \bar{I}_0 + \dot{\beta}_s Y_r$, amortized over the index set $\mathcal{T}$:

$$\mathcal{L}_{\text{diag}}(\hat{v}) := \int_{\mathcal{T}} \int_{\mathcal{T}} \int_0^1 \mathbb{E}\left[\left|\hat{v}_{s,s}(\bar{I}_s; r, t, Y_t) - \tfrac{d}{ds}\bar{I}_s\right|^2\right] ds \, d\mu(t) \, d\mu(r), \tag{49}$$

where $\mu$ is a measure on $\mathcal{T}$ (such as the uniform distribution). Minimizing $\mathcal{L}_{\text{diag}}(\hat{v})$ ensures that

$$\hat{v}_{s,s}(\bar{x}; r, t, x) = \mathbb{E}\left[\tfrac{d}{ds}\bar{I}_s \mid \bar{I}_s = \bar{x}, X_t = x\right] = \mathbb{E}\left[\dot{\alpha}_s \bar{I}_0 + \dot{\beta}_s X_r \mid \bar{I}_s = \bar{x}, X_t = x\right] = \bar{b}_s(\bar{x}; r, t, x). \tag{50}$$

Consistency is enforced by applying any standard consistency objective, such as those in Table 2, to the conditional map $\hat{X}_{s,u}$, where the loss is augmented by passing the target time $r$ and conditioning state $(t, X_t)$ as additional inputs to the network.

### E.7. Extended MFM-FT

Define the conditional posteriors

$$p_{r|t}(\cdot|x) := \text{Law}(X_r|X_t = x). \tag{51}$$

Suppose that for some base distribution $q$, the MFM $X_{s,u}(\cdot\,; r, t, x)$ satisfies

$$X_{0,1}(\cdot\,; r, t, x)\#q = p_{r|t}(\cdot|x), \qquad \forall r, t \in [0,1], \forall x \in \mathbb{R}^d. \tag{52}$$

To ease notation, we define

$$\Phi(\cdot\,; r, t, x) := X_{0,1}(\cdot\,; r, t, x). \tag{53}$$

Notice that

$$\nabla V_t(x) = \nabla \log \mathbb{E}[e^{r(X_1)}|X_t = x] \tag{54}$$

$$= \frac{\nabla \mathbb{E}[e^{r(X_1)}|X_t = x]}{\mathbb{E}[e^{r(X_1)}|X_t = x]} \tag{55}$$

$$= \frac{\nabla \mathbb{E}\left[\mathbb{E}[e^{r(X_1)}|X_r]|X_t = x\right]}{\mathbb{E}[e^{r(X_1)}|X_t = x]} \tag{56}$$

$$= \frac{\nabla \mathbb{E}\left[\exp\left(r\left(\Phi(\epsilon_2; 1, r, \Phi(\epsilon_1; r, t, x))\right)\right)\right]}{\mathbb{E}[e^{r(X_1)}|X_t = x]}, \tag{57}$$

where $\epsilon_1, \epsilon_2 \overset{\text{iid}}{\sim} q$. Let $\hat{x}_r = \Phi(\epsilon_1; r, t, x)$ and $\hat{x}_1 = \Phi(\epsilon_2; 1, r, \hat{x}_r)$. Then

$$\nabla V_t(x) = \frac{\mathbb{E}\left[\exp\left(r(\hat{x}_1)\right)\nabla r(\hat{x}_1)J\Phi(\epsilon_2; 1, r, \hat{x}_r)J\Phi(\epsilon_1; r, t, x)\right]}{\mathbb{E}[e^{r(X_1)}|X_t = x]}$$

$$= \frac{\mathbb{E}\left[\nabla V_r(\hat{x}_r)\mathbb{E}\left[e^{r(X_1)}|X_r = \hat{x}_r\right]J\Phi(\epsilon_1; r, t, x)\right]}{\mathbb{E}[e^{r(X_1)}|X_t = x]}.$$

Therefore

$$\mathbb{E}[e^{r(X_1)}|X_t = x]\nabla V_t(x) = \mathbb{E}\left[\nabla V_r(\hat{x}_r)\mathbb{E}\left[e^{r(X_1)}|X_r = \hat{x}_r\right]J\Phi(\epsilon_1; r, t, x)\right].$$

### E.8. MFM-Search

In settings where the reward function, $r(x)$, is non-differentiable, the highly-performant gradient estimator presented in (14) can not be used. Although MFM-GF (32) can be used in such settings, it scales poorly to high-dimensional problems. As such, we present an alternate search-based algorithm for reward maximisation, which we call MFM-Search.

---

**Algorithm 1** MFM Search (Gradient-Free)

---

**Input:** Reward $r(x)$; MFM $X$; schedule $0 = t_0 < \cdots < t_K = 1$; number of candidates $m$; posterior samples per candidate $N$

Initialize candidate set $\mathcal{C}_0 = \{x_{t_0}^{(m)}\}_{m=1}^M$ with $x_{t_0}^{(m)} \sim p_0$

**for** $k = 0$ **to** $K - 1$ **do**

    # Sample posterior candidates via 1-step MFM sampler

    **for** $m = 1$ **to** $M$ **do**

        **for** $n = 1$ **to** $N$ **do**

            Sample $\epsilon^{(m,n)} \sim p_0$

            $\hat{x}_1^{(m,n)} \leftarrow X_{t_k,1}(\epsilon^{(m,n)}, x_{t_k}^{(m)})$

        **end for**

    **end for**

    # Select best candidate across all projections

    $(m^\star, n^\star) \leftarrow \arg\max\limits_{m,n} r\left(\hat{x}_1^{(m,n)}\right)$

    $x_1^\star \leftarrow \hat{x}_1^{(m^\star, n^\star)}$

    # Re-noise best candidate to form next candidate set

    **for** $m = 1$ **to** $M$ **do**

        Sample $\tilde{\epsilon}^{(m)} \sim p_0$

        $x_{t_{k+1}}^{(m)} \leftarrow \beta_{t_{k+1}} x_1^\star + \alpha_{t_{k+1}} \tilde{\epsilon}^{(m)}$

    **end for**

    $\mathcal{C}_{k+1} \leftarrow \{x_{t_{k+1}}^{(m)}\}_{m=1}^M$

**end for**

**Output:** Selected sample $x_1^\star$

---

We emphasize that MFM-Search does not provide asymptotic guarantees of sampling the true tilted distribution, $p_{\text{reward}} \propto p_{\text{model}}(x)e^{r(x)}$. Instead, it should be viewed as a heuristic search procedure that exploits the efficiency of posterior sampling through MFMs to efficiently explore high-reward regions of the sample space. Note that Algorithm 1 is a single instantiation within a broader design space of search and optimisation algorithms enabled by MFMs, and we leave a systematic exploration of this space to future work.

## F. Consistency Objectives

### F.1. Flow Map Objectives

The consistency rules in Equation (9) can directly be enforced by penalizing the residual violations of the rules for all $s, u, w \in [0, 1]$ and $x \in \mathbb{R}^d$:

$$\left\|\partial_u \hat{X}_{s,u}(x) - \hat{v}_{u,u}\big(\hat{X}_{s,u}(x)\big)\right\|^2 \tag{58}$$

$$\left\|\partial_s \hat{X}_{s,u}(x) + \hat{v}_{s,s}(x) \cdot \nabla \hat{X}_{s,u}(x)\right\|^2 \tag{59}$$

$$\left\|\hat{X}_{w,u}\big(\hat{X}_{s,w}(x)\big) - \hat{X}_{s,u}(x)\right\|^2 \tag{60}$$

### F.2. MFM Objectives

As noted in the main body, any consistency objective used in the training of standard flow-maps can be adapted for MFM training. We present four representative examples in Table 2.

*Table 2.* MFM consistency objectives. We adapt the Eulerian, Lagrangian, and Semigroup losses (Boffi et al., 2025a) and the Mean Flow loss (Geng et al., 2025a) to the MFM framework. We distinguish between two supervision variants: teacher-distillation, which leverages the analytical drift $\bar{b}_s$ of a pretrained flow matching model $b_t$ for stable supervision, and self-distillation, where the model bootstraps its own targets when training directly from data. The operator $\text{sg}(\cdot)$ signifies that we place a stop gradient on the term during optimization.

| Objective | Distillation | Loss Formulation $\mathcal{L}_{\text{cons}}(\hat{v})$ |
|---|---|---|
| **Eulerian** | Self | $\int_0^1 \int_0^u \mathbb{E} \left\| \hat{v}_{s,u}(I_s; t, I_t') - \text{sg}\Big((u-s)\partial_s \hat{v}_{s,u}(I_s; t, I_t') + v_{s,s}(I_s; t, I_t') \cdot \nabla \hat{X}_{s,u}(I_s; t, I_t')\Big) \right\|^2 ds du$ |
| | Teacher | $\int_0^1 \int_0^u \mathbb{E} \left\| \hat{v}_{s,u}(I_s; t, I_t') - \text{sg}\Big((u-s)\partial_s \hat{v}_{s,u}(I_s; t, I_t') + \bar{b}_s(I_s; t, I_t') \cdot \nabla \hat{X}_{s,u}(I_s; t, I_t')\Big) \right\|^2 ds du$ |
| **Lagrangian** | Self | $\int_0^1 \int_0^u \mathbb{E} \left\| \hat{v}_{s,u}(I_s; t, I_t') - \text{sg}\Big(\hat{v}_{u,u}\big(\hat{X}_{s,u}(I_s; t, I_t'); t, I_t'\big) - (u-s)\partial_u \hat{v}_{s,u}(I_s; t, I_t')\Big) \right\|^2 ds du$ |
| | Teacher | $\int_0^1 \int_0^u \mathbb{E} \left\| \hat{v}_{s,u}(I_s; t, I_t') - \text{sg}\Big(\bar{b}_u\big(\hat{X}_{s,u}(I_s; t, I_t'); t, I_t'\big) - (u-s)\partial_u \hat{v}_{s,u}(I_s; t, I_t')\Big) \right\|^2 ds du$ |
| **Mean Flow** | Self | $\int_0^1 \int_0^u \mathbb{E} \left\| \hat{v}_{s,u}(I_s; t, I_t') - \text{sg}\Big((u-s)\partial_s \hat{v}_{s,u}(I_s; t, I_t') + \dot{I}_s \cdot \nabla \hat{X}_{s,u}(I_s; t, I_t')\Big) \right\|^2 ds du$ |
| **Semigroup** | Self | $\int_0^1 \int_0^u \int_s^u \mathbb{E} \left\| \hat{X}_{s,u}(I_s; t, I_t') - \text{sg}\Big(\hat{X}_{w,u}(\hat{X}_{s,w}(I_s; t, I_t'); t, I_t')\Big) \right\|^2 dw ds du$ |

# G. Implementation Details

## G.1. Model Guidance

Classifier-Free Guidance (CFG) interpolates between conditional and unconditional velocities at inference-time, requiring two function evaluations for every denoising step (Ho & Salimans, 2022):

$$\tilde{v}_\theta(x_t, t, y) = v_\theta(x_t, t, y) + \omega_{CFG} \cdot (v_\theta(x_t, t, y) - v_\theta(x_t, t, \emptyset)) \tag{61}$$

Tang et al. (2025) recently proposed Model Guidance (MG) to learn this interpolated velocity during training, in order to reduce the inference cost from two to one model evaluation per denoising step. This approach has been shown to be particularly effective for achieving competitive one and few-step performance (Geng et al., 2025a; Lee et al., 2025). The standard training objective for MG replaces the base class-conditioned drift with a target that leverages the model's current conditional and unconditional drift estimates, as well as the class-conditioned velocity from data:

$$v^{\text{tgt}}(I_t, t, y) = \dot{I}_t + \omega \cdot \text{sg}(v_\theta(I_t, t, y) - v_\theta(I_t, t, \emptyset)) \tag{62}$$

where $\omega \in (0, 1)$ is the model guidance scale. The stop-gradient operator, denoted as $\text{sg}(\cdot)$, ensures training stability. By considering the fixed point of this objective, it is easy to show that this training target is equivalent to using a standard CFG scale of $\omega_{CFG} = 1/(1 - \omega)$ during sampling.

For MFMs, we extend MG by conditioning on an arbitrary pair $(t, x)$ along the stochastic interpolant. To this end, aligning with notation in Equation 20, the MFM training target becomes:

$$v^{\text{tgt}}_{s,s}(\bar{I}_s; t, I_t, y) = \frac{d}{ds}\bar{I}_s + \omega \cdot \text{sg}\big(\hat{v}_{s,s}(\bar{I}_s; t, I_t, y) - \hat{v}_{s,s}(\bar{I}_s; t, I_t, \emptyset)\big) \tag{63}$$

Recent works also consider amortizing over a range of $\omega$ values to allow further flexibility at inference time by passing an additional input $\omega$ to the network (Geng et al., 2025b). Through minimizing this objective on the diagonal, alongside a consistency objective of choice, we enable 1-NFE generation that recovers the desired CFG/MG distribution for all $(t, x)$. We leverage MG for training MFM models on ImageNet. Parameter choices are specified in Table 5.

## G.2. Architecture

We leverage standard diffusion and flow architectures (Karras et al., 2022; Peebles & Xie, 2023) with extensions to accommodate the additional network inputs. For our ImageNet experiments, we use Diffusion Transformer (DiT) backbone following related works (Geng et al., 2025a; Lee et al., 2025). While we retain the core transformer blocks, we introduce

two specific embedding mechanisms to condition standard architectures for flow maps, $\hat{X}_{s,u}(.)$, on the outer flow time $t$ and the corresponding state $x$, to yield an MFM, $\hat{X}_{s,u}(.; t, x)$

In DiT-based flow maps, a global conditioning vector $c$, which is a function of start and endpoint times, $s, u$, and any additional conditioning signals, e.g. class, is used throughout the network. For MFMs, we extend this vector to also incorporate outer-time $t$. In order to condition on the spatial state $x$, we introduce an additional patch embedder layer. We then form the input to the DiT as a combination of the inner-state, $x_s$, and outer conditioning state, $x$ modulated by AdaLN-Zero conditioned on $t$:

$$c = \underbrace{\text{Embed}_{\text{time},s}(s) + \text{Embed}_{\text{time},u}(u) + \text{Embed}_{\text{class}}(y)}_{\text{Flow map}} + \underbrace{\text{Embed}_{\text{time},t}(t)}_{\textbf{MFM}} \tag{64}$$

$$x_{\text{input}} = \underbrace{\text{PatchEmbed}(x_s) + \text{PosEmbed}}_{\text{Flow map}} + \underbrace{\text{AdaLN-Zero}[\text{PatchEmbed}'(x)|t]}_{\textbf{MFM}} \tag{65}$$

**Fine-Tuning from a Flow map**  For fine-tuning from a flow map, the flow map can be preserved at initialisation through zero initialisation of the $t$ time embedding MLP, and ensuring the new AdaLN-Zero modulates the contribution of $x$ to an all zero tensor.

**Inductive Bias at $t = 0$**  As highlighted in Section 4.3, $p_{1|0}(.|x_0) = p_1$ for any $x_0$, meaning the conditional velocities and maps should in fact be independent of $x_t$ at $t = 0$. As such, architectures that ensure that $x$ is ignored by design at $t = 0$ (and diminished at low $t$) can be considered to improve performance.

The limited parameter and architectural overhead allows MFMs to be implemented into popular generative modelling workflows and codebases.

### G.3. Adaptive Loss

In our ImageNet experiments, we follow Mean Flow (Geng et al., 2025a) and use an adaptive loss for both the diagonal and consistency terms of the MFM loss. The adaptively weighted loss is $\text{sg}(w) \cdot \mathcal{L}$, with $\mathcal{L} = \|\Delta\|_2^2$, where $\|\Delta\|$ denotes the regression error. The weights are set as follows:

$$w = \frac{1}{(\|\Delta\|_2^2 + c)^p}, \tag{66}$$

where $c > 0$ and $p > 0$ are hyper-parameters. Note that $p = 0, c = 0$ recovers the standard $\mathcal{L}_2$ loss. See Table 4 for further results on different choices of these parameters.

## H. Proofs

### H.1. Conditional Endpoint Law

**Proposition H.1.** *Assume $I_0 \sim p_0 = \mathcal{N}(0, I)$ and consider the interpolant*

$$I_t = \alpha_t I_0 + \beta_t I_1, \tag{67}$$

*with $\alpha_0 = \beta_1 = 1$ and $\alpha_1 = \beta_0 = 0$. Define*

$$\frac{\sigma_t^2}{2} = \frac{\dot{\beta}_t}{\beta_t}\alpha_t^2 - \dot{\alpha}_t\alpha_t, \tag{68}$$

*and let $p_t = \text{Law}(I_t)$. Consider the SDE*

$$dX_t = f_t(X_t)dt + \sigma_t dB_t, \qquad f_t(x) = b_t(x) + \frac{\sigma_t^2}{2}\nabla \log p_t(x), \qquad X_0 \sim p_0, \tag{69}$$

*where*

$$b_t(x) = \mathbb{E}[\dot{I}_t | I_t = x]. \tag{70}$$

*Then for all $t \in [0, 1]$ and $x \in \mathbb{R}^d$*

$$\text{Law}(X_1 | X_t = x) = \text{Law}(I_1 | I_t = x). \tag{71}$$

*Proof.* Note that $\text{Law}(X_t) = p_t$, see (Song et al., 2020; Albergo et al., 2023b). Therefore the time reversal $(Y_t)_{t \in [0,1]} := (X_{1-t})_{t \in [0,1]}$ satisfies the following SDE (Anderson, 1982)

$$dY_t = \tilde{f}_{1-t}(Y_t) + \sigma_{1-t}dB_t, \qquad \tilde{f}_t(x) = -b_t(x) + \frac{\sigma_t^2}{2}\nabla \log p_t(x), \qquad Y_0 \sim p_1. \tag{72}$$

By Tweedie's formula we have the identity

$$\nabla \log p_t(x) = \frac{-x + \beta_t \mathbb{E}[I_1 | I_t = x]}{\alpha_t^2}. \tag{73}$$

This gives

$$\tilde{f}_t(x) = -\dot{\alpha}_t \mathbb{E}[I_0 | I_t = x] - \dot{\beta}_t \mathbb{E}[I_1 | I_t = x] + (\frac{\dot{\beta}_t}{\beta_t}\alpha_t^2 - \dot{\alpha}_t \alpha_t)\frac{-x + \beta_t \mathbb{E}[I_1 | I_t = x]}{\alpha_t^2} \tag{74}$$

$$= -\dot{\alpha}_t \mathbb{E}[I_0 | I_t = x] - \frac{\dot{\alpha}_t}{\alpha_t}\beta_t \mathbb{E}[I_1 | I_t = x] + (\frac{\dot{\alpha}_t}{\alpha_t} - \frac{\dot{\beta}_t}{\beta_t})x. \tag{75}$$

Using the relation $x = \alpha_t \mathbb{E}[I_0 | I_t = x] + \beta_t \mathbb{E}[I_1 | I_t = x]$ we obtain

$$\tilde{f}_t(x) = -\frac{\dot{\beta}_t}{\beta_t}x. \tag{76}$$

We use an integrating factor. Let $\Phi(t) = \int_0^t -\frac{\dot{\beta}_{1-s}}{\beta_{1-s}}ds = \log \beta_{1-t}$ and by Ito's formula we have

$$d\left(e^{-\Phi(t)}Y_t\right) = -\Phi'(t)e^{-\Phi(t)}Y_t dt + e^{-\Phi(t)}dY_t \tag{77}$$

$$= \frac{\dot{\beta}_{1-t}}{\beta_{1-t}}e^{-\Phi(t)}Y_t dt - e^{-\Phi(t)}\frac{\dot{\beta}_{1-t}}{\beta_{1-t}}Y_t dt + e^{-\Phi(t)}\sigma_{1-t}dB_t \tag{78}$$

$$= e^{-\Phi(t)}\sigma_{1-t}dB_t. \tag{79}$$

Therefore

$$Y_t = e^{\Phi(t)}Y_0 + e^{\Phi(t)}\int_0^t e^{-\Phi(s)}\sigma_{1-s}dB_s. \tag{80}$$

Notice that $\int_0^t e^{-\Phi(s)}\sigma_{1-s}dB_s$ is a mean zero Gaussian with variance equal to

$$\int_0^t e^{-2\phi(s)}\sigma_{1-s}^2 ds = \int_0^t 2\beta_{1-s}^{-2}(\frac{\dot{\beta}_{1-s}}{\beta_{1-s}}\alpha_{1-s}^2 - \dot{\alpha}_{1-s}\alpha_{1-s})ds \tag{81}$$

$$= \int_0^t \frac{d}{ds}\left(\frac{\alpha_{1-s}^2}{\beta_{1-s}^2}\right)ds \tag{82}$$

$$= \frac{\alpha_{1-t}^2}{\beta_{1-t}^2}. \tag{83}$$

Therefore

$$\text{Law}(Y_t | Y_0) = \text{Law}(\alpha_{1-t}Z + \beta_{1-t}Y_0 | Y_0), \tag{84}$$

for some $Z \sim \mathcal{N}(0, I)$ independent of $Y_0$. Since $Y$ is the time reversal of $X$, we have for all $t \in [0,1]$ and $x \in \mathbb{R}^d$

$$\text{Law}(X_t | X_1 = x) = \text{Law}(\alpha_t Z + \beta_t I_1 | x), \tag{85}$$

where $Z \sim \mathcal{N}(0, I)$ is independent of $I_1$. Moreover, we also have equality of the marginal laws $\text{Law}(X_1) = \text{Law}(I_1)$, so the joint laws of the SDE and the interpolant coincide:

$$\text{Law}(X_t, X_1) = \text{Law}(I_t, I_1). \tag{86}$$

Equality of the joint laws implies equality of the corresponding conditional laws for all $t \in [0,1]$ and $x \in \mathbb{R}^d$

$$\text{Law}(X_1 | X_t = x) = \text{Law}(I_1 | I_t = x), \tag{87}$$

which is precisely the desired claim. □

## H.2. Convergence Guarantees

In this section, we provide a proof for the convergence rates stated in Proposition H.2.

**Proposition H.2** (Formal Convergence Guarantees). *Let $p_{\text{reward}}$ denote the target distribution defined in (1). Let $\hat{p}_1$ denote the terminal distribution generated by the MFM steering (SDE) sampler (Algorithm 4) using a uniform time discretization with $K$ steps ($t_k = k/K$) and $N$ independent Monte Carlo samples per step.*

*Suppose the following regularity conditions hold:*

1. *The reward function $r \in C^1(\mathbb{R}^d)$ and its gradient $\nabla r$ are both bounded.*

2. *The MFM $f(\epsilon, t, x) \in C^1(\mathbb{R}^d)$ in $x$ and $\nabla_x f$ is bounded.*

3. *The optimal drift $b_t^\star(x)$ is $L$-Lipschitz continuous in space and $1/2$-Hölder continuous in time. That is, for all $t, s \in [0,1]$ and $x, y \in \mathbb{R}^d$:*

$$|b_t^\star(x) - b_t^\star(y)| \le L|x-y| \quad and \quad |b_t^\star(x) - b_s^\star(x)| \le C_{\text{time}}|t-s|^{1/2}. \tag{88}$$

4. *There exist $\sigma_{\max} > \sigma_{\min} > 0$ such that $\sigma_{\min} \le \sigma_t \le \sigma_{\max}$ for all $t \in [0,1]$.*

*Then, there exists a constant $C > 0$ independent of $K$ and $N$ such that the convergence to the target satisfies:*

$$W_2(\hat{p}_1, p_{\text{reward}}) \le C\left(\frac{1}{\sqrt{K}} + \frac{1}{N}\right) \quad and \quad \text{KL}(\hat{p}_1 \| p_{\text{reward}}) \le C\left(\frac{1}{K} + \frac{1}{N}\right). \tag{89}$$

*Remark* H.3. We note that the regularity conditions on the drift may be violated at $t=1$ if the target distribution is supported on a low-dimensional manifold, as the score function becomes singular. Following standard practice in diffusion theory, our results formally apply to the process stopped at $t = 1 - \varepsilon$ for a small $\varepsilon > 0$, where the score is smooth and the Lipschitz constant $L_\varepsilon$ is finite. In the case of the Wasserstein bound, we pick up an additional term $W_2(p_{1-\varepsilon}, p_{\text{reward}})$ corresponding to the smoothing error, which allows us to bound the distance to the exact target.

*Proof.* To analyze the convergence, we view the discrete sampling algorithm (Algorithm 4) as a continuous-time randomized Euler scheme by interpolating the discrete algorithm. Let $t_k = k/K$ for $k = 0, \ldots, K$ be a uniform time grid with step size $\delta = 1/K$. We define the randomized interpolant $\widehat{X}_t$ as the continuous process satisfying:

$$d\widehat{X}_t = \hat{b}_{\eta(t)}^{(N)}(\widehat{X}_{\eta(t)})dt + \sigma_t dB_t, \quad \widehat{X}_0 \sim p_0, \tag{90}$$

where $\eta(t) = t_k$ for $t \in [t_k, t_{k+1})$. The drift $\hat{b}_{t_k}^{(N)}(x)$ is the Monte Carlo estimator of the optimal drift derived using $N$ independent samples. Crucially, for each step $k$, we draw a fresh batch of $N$ samples $\epsilon^{(k,1)}, \ldots, \epsilon^{(k,N)} \sim p_0$ to construct the estimator. We use the Gradient Estimator defined in (14) for the drift. Specifically, $\hat{b}_t^{(N)}(x)$ estimates the optimal drift $b_t^\star(x) = b_t(x) + \frac{\sigma_t^2}{2}\nabla \log p_t(x) + \sigma_t^2 \nabla V_t(x)$. Using the reparameterization $f(\epsilon, t, x) = \Phi_{0,1}(\epsilon; t, x)$, the gradient estimator is:

$$\hat{b}_t^{(N)}(x) = b_t(x) + \frac{\sigma_t^2}{2}\nabla \log p_t(x) + \sigma_t^2 \nabla_x \log\left(\frac{1}{N}\sum_{i=1}^{N} e^{r(f(\epsilon^{(i)}, t, x))}\right). \tag{91}$$

We compare $\widehat{X}_t$ to the optimal steered process $X_t^\star$ governed by the exact drift $b_t^\star(x)$:

$$dX_t^\star = b_t^\star(X_t^\star)dt + \sigma_t dB_t, \quad X_0^\star \sim p_0. \tag{92}$$

Our main analysis relies on controlling the error of the Monte Carlo drift estimator. The following proposition establishes the bias and variance bounds that will be central to our main proof.

**Proposition H.4** (Drift Estimator Moments). *There exists a constant $C'$ such that for any $x$:*

$$\underbrace{\left|\mathbb{E}[\hat{b}_t^{(N)}(x)] - b_t^\star(x)\right|}_{Bias} \le \frac{C'}{N}, \qquad \underbrace{\mathbb{E}\left[\left|\hat{b}_t^{(N)}(x) - \mathbb{E}[\hat{b}_t^{(N)}(x)]\right|^2\right]}_{Variance} \le \frac{C'}{N}. \tag{93}$$

*Proof.* Recall that the estimator is given by $\hat{b}_t^{(N)}(x) = b_t(x) + \frac{\sigma_t^2}{2}\nabla \log p_t(x) + \sigma_t^2 \widehat{\nabla V}_t(x)$, where $b_t(x) + \frac{\sigma_t^2}{2}\nabla \log p_t(x)$ is deterministic. Therefore, the bias and variance of $\hat{b}_t^{(N)}(x)$ are determined by the properties of the gradient estimator $\widehat{\nabla V}_t(x)$. Recall that $\nabla V_t(x) = \frac{\mathbb{E}[e^{r(f(\epsilon,t,x))}\nabla_x(r\circ f)(\epsilon,t,x)]}{\mathbb{E}[e^{r(f(\epsilon,t,x))}]}$. The estimator takes the form of a ratio of sample means $\frac{\bar{G}}{\bar{W}} = \frac{\frac{1}{N}\sum G_i}{\frac{1}{N}\sum W_i}$, where $W_i = e^{r(f(\epsilon^{(i)},t,x))}$ and $G_i = W_i\nabla(r\circ f)(\epsilon^{(i)},t,x)$. Let $\mu_W = \mathbb{E}[W_i]$ and $\mu_G = \mathbb{E}[G_i]$. The target is $\frac{\mu_G}{\mu_W}$. To analyze the bias and variance, we apply the multivariate Taylor expansion of the function $h(u,v) = u/v$ around the point $(\mu_G, \mu_W)$. The partial derivatives evaluated at the mean are:

$$\frac{\partial h}{\partial u} = \frac{1}{\mu_W}, \quad \frac{\partial h}{\partial v} = -\frac{\mu_G}{\mu_W^2}, \quad \frac{\partial^2 h}{\partial u^2} = 0, \quad \frac{\partial^2 h}{\partial v^2} = \frac{2\mu_G}{\mu_W^3}, \quad \frac{\partial^2 h}{\partial u \partial v} = -\frac{1}{\mu_W^2}. \tag{94}$$

**Bias Analysis.** Expanding $h(\bar{G}, \bar{W}) = \frac{\bar{G}}{\bar{W}}$ to the second order yields:

$$\frac{\bar{G}}{\bar{W}} = \frac{\mu_G}{\mu_W} + \frac{1}{\mu_W}(\bar{G} - \mu_G) - \frac{\mu_G}{\mu_W^2}(\bar{W} - \mu_W) \tag{95}$$

$$+ \frac{1}{2}\left[\frac{2\mu_G}{\mu_W^3}(\bar{W} - \mu_W)^2 - \frac{2}{\mu_W^2}(\bar{G} - \mu_G)(\bar{W} - \mu_W)\right] + R_3, \tag{96}$$

where $R_3$ is the remainder term. Taking the expectation, the first-order terms vanish since $\mathbb{E}[\bar{G}] = \mu_G$ and $\mathbb{E}[\bar{W}] = \mu_W$. For the second-order terms, we utilize the properties of the sample mean variances and covariances:

$$\mathbb{E}[(\bar{W} - \mu_W)^2] = \frac{1}{N}\text{Var}(W_i), \quad \mathbb{E}[(\bar{G} - \mu_G)(\bar{W} - \mu_W)] = \frac{1}{N}\text{Cov}(G_i, W_i). \tag{97}$$

Substituting these into the expectation:

$$\mathbb{E}\left[\frac{\bar{G}}{\bar{W}}\right] - \frac{\mu_G}{\mu_W} = \frac{1}{N}\left(\frac{\mu_G}{\mu_W^3}\text{Var}(W_i) - \frac{1}{\mu_W^2}\text{Cov}(G_i, W_i)\right) + \mathbb{E}[R_3]. \tag{98}$$

By our boundedness assumptions, $W_i$ and $G_i$ have bounded moments, and $\mu_W > 0$. The expectation of the remainder $\mathbb{E}[R_3]$ is dominated by third-order central moments of the sample means, which scale as $\mathcal{O}(N^{-2})$ for i.i.d. variables with bounded moments. Thus, the bias is dominated by the $1/N$ term:

$$\left|\mathbb{E}[\hat{b}_t^{(N)}(x)] - b_t^\star(x)\right| \leq \frac{C'}{N}. \tag{99}$$

**Variance Analysis.** Using the second-order Taylor expansion of $h$ around $(\mu_G, \mu_W)$, we have:

$$\frac{\bar{G}}{\bar{W}} = \frac{\mu_G}{\mu_W} + \mathcal{L} + R_2, \tag{100}$$

where $\mathcal{L} = \frac{1}{\mu_W}(\bar{G} - \mu_G) - \frac{\mu_G}{\mu_W^2}(\bar{W} - \mu_W)$ is the first-order linear term and $R_2$ is the remainder. Since $\mathbb{E}[\mathcal{L}] = 0$, the expectation of the estimator is

$$\mathbb{E}\left[\frac{\bar{G}}{\bar{W}}\right] = \frac{\mu_G}{\mu_W} + \mathbb{E}[R_2]. \tag{101}$$

Substituting this into the variance definition:

$$\text{Var}\left(\frac{\bar{G}}{\bar{W}}\right) = \mathbb{E}\left[\left((\frac{\mu_G}{\mu_W} + \mathcal{L} + R_2) - (\frac{\mu_G}{\mu_W} + \mathbb{E}[R_2])\right)^2\right] \tag{102}$$

$$= \mathbb{E}\left[(\mathcal{L} + (R_2 - \mathbb{E}[R_2]))^2\right] \tag{103}$$

$$= \mathbb{E}[\mathcal{L}^2] + 2\mathbb{E}[\mathcal{L}(R_2 - \mathbb{E}[R_2])] + \text{Var}(R_2). \tag{104}$$

The dominant term is the variance of the linear approximation $\mathbb{E}[\mathcal{L}^2]$:

$$\mathbb{E}[\mathcal{L}^2] = \frac{1}{N}\left(\frac{\text{Var}(G_i)}{\mu_W^2} + \frac{\mu_G^2\text{Var}(W_i)}{\mu_W^4} - \frac{2\mu_G\text{Cov}(G_i, W_i)}{\mu_W^3}\right). \tag{105}$$

The other terms involve the remainder $R_2$, which scales as $\mathcal{O}(N^{-1})$. Therefore, the cross-term $\mathbb{E}[\mathcal{L}R_2]$ and $\mathrm{Var}(R_2)$ scale as $\mathcal{O}(N^{-2})$. Given the boundedness assumptions on $r$ and its gradients, the moments of $G_i$ and $W_i$ are finite. Thus, we obtain the bound:

$$\mathrm{Var}\left(\hat{b}_t^{(N)}(x)\right) = \sigma_t^4 \mathrm{Var}\left(\frac{\bar{G}}{\bar{W}}\right) \le \frac{C'}{N}. \tag{106}$$

$\square$

We return to our main proof. We use the notation $\lesssim$ to denote inequality up to a multiplicative constant independent of $N$ and $K$, simplifying the presentation by suppressing non-essential factors.

**Wasserstein-2 Bound.** We use a synchronous coupling where both processes are driven by the same Brownian motion $B_t$. Let $e_t = \widehat{X}_t - X_t^\star$. The error evolves according to:

$$\frac{d}{dt} e_t = \hat{b}_{\eta(t)}^{(N)}(\widehat{X}_{\eta(t)}) - b_t^\star(X_t^\star). \tag{107}$$

We decompose the drift mismatch as follows:

$$\hat{b}_{\eta(t)}^{(N)}(\widehat{X}_{\eta(t)}) - b_t^\star(X_t^\star) = \underbrace{[\hat{b}_{\eta(t)}^{(N)}(\widehat{X}_{\eta(t)}) - \bar{b}_{\eta(t)}(\widehat{X}_{\eta(t)})]}_{\xi_t} + \underbrace{[\bar{b}_{\eta(t)}(\widehat{X}_{\eta(t)}) - b_{\eta(t)}^\star(\widehat{X}_{\eta(t)})]}_{\Delta_t} + \underbrace{[b_{\eta(t)}^\star(\widehat{X}_{\eta(t)}) - b_t^\star(X_t^\star)]}_{D_t}, \tag{108}$$

where $\bar{b}_t(x) = \mathbb{E}[\hat{b}_t^{(N)}(x)]$ denotes the expectation over the random samples $\{\epsilon^{(i)}\}_{i=1}^N$. Integrating and taking expectations yields:

$$\mathbb{E}|e_t|^2 \lesssim \mathbb{E}\left|\int_0^t \xi_s ds\right|^2 + \mathbb{E}\left|\int_0^t \Delta_s ds\right|^2 + \mathbb{E}\left|\int_0^t D_s ds\right|^2. \tag{109}$$

Next, we bound each of the three terms separately.

**Martingale Term ($\xi_t$).** Since independent samples are used for each interval $[t_k, t_{k+1}]$, the integral represents a sum of martingale increments. By orthogonality:

$$\mathbb{E}\left|\int_0^t \xi_s ds\right|^2 \le \sum_{k=0}^{K-1} \mathbb{E}\left|\int_{t_k}^{t_{k+1}} \xi_s ds\right|^2 = \sum_{k=0}^{K-1} \delta^2 \mathbb{E}|\xi_{t_k}|^2. \tag{110}$$

Using the variance bound from Proposition H.4 ($\mathbb{E}\|\xi_{t_k}\|^2 \lesssim N^{-1}$), we have:

$$\mathbb{E}\left|\int_0^t \xi_s ds\right|^2 \lesssim \sum_{k=0}^{K-1} \delta^2 \frac{1}{N} = \frac{1}{NK}. \tag{111}$$

**Bias Term ($\Delta_t$).** Using Jensen's inequality and the bias bound from Proposition H.4,

$$\mathbb{E}\left\|\int_0^t \Delta_s ds\right\|^2 \le t \int_0^t \mathbb{E}\|\Delta_s\|^2 ds \lesssim \frac{1}{N^2}. \tag{112}$$

**Discretization Term ($D_t$).** We assume $b^\star$ is $L$-Lipschitz in space and $1/2$-Hölder continuous in time. Using the triangle inequality:

$$|D_s| \le |b_{\eta(s)}^\star(\widehat{X}_{\eta(s)}) - b_s^\star(\widehat{X}_{\eta(s)})| + |b_s^\star(\widehat{X}_{\eta(s)}) - b_s^\star(\widehat{X}_s)| + |b_s^\star(\widehat{X}_s) - b_s^\star(X_s^\star)| \tag{113}$$

$$\lesssim \sqrt{|s - \eta(s)|} + |\widehat{X}_{\eta(s)} - \widehat{X}_s| + |e_s|. \tag{114}$$

Integrating and taking expectation implies:

$$\mathbb{E}\left|\int_0^t D_s ds\right|^2 \lesssim \int_0^t \left(\delta + \mathbb{E}|\widehat{X}_{\eta(s)} - \widehat{X}_s|^2 + \mathbb{E}|e_s|^2\right) ds. \tag{115}$$

To handle the term $\mathbb{E}|\widehat{X}_{\eta(s)} - \widehat{X}_s|^2$, recall that $\widehat{X}_s - \widehat{X}_{\eta(s)} = \int_{\eta(s)}^s \hat{b}_u du + \int_{\eta(s)}^s \sigma_u dB_u$. The squared expectation is dominated by the diffusion term (via Itô isometry), which scales as the interval length $\delta$, whereas the drift term scales as $\delta^2$. Thus, we have the standard Euler-Maruyama bound $\mathbb{E}|\widehat{X}_{\eta(s)} - \widehat{X}_s|^2 \lesssim \delta$. Substituting this back:

$$\mathbb{E}\left|\int_0^t D_s ds\right|^2 \lesssim \delta + \int_0^t \mathbb{E}|e_s|^2 ds. \tag{116}$$

**Completing the Bound.** Combining the terms leads to:

$$\mathbb{E}|e_t|^2 \lesssim \frac{1}{NK} + \frac{1}{N^2} + \frac{1}{K} + \int_0^t \mathbb{E}|e_s|^2 ds. \tag{117}$$

Applying Grönwall's lemma:

$$\sup_{t \in [0,1]} \mathbb{E}|e_t|^2 \lesssim \frac{1}{NK} + \frac{1}{N^2} + \frac{1}{K}. \tag{118}$$

Taking square root and only keeping the dominant terms yields the Wasserstein-2 bound:

$$W_2(\hat{p}_1, p_{\text{reward}}) \leq \sqrt{\mathbb{E}|e_1|^2} \lesssim \frac{1}{\sqrt{K}} + \frac{1}{N}. \tag{119}$$

**KL Divergence Bound.** We employ the data-processing inequality to bound the divergence between the sampling distribution and the target by the divergence between their path measures: $\text{KL}(\hat{p}_1 \| p_{\text{reward}}) \leq \text{KL}(\mathbb{P}^{\widehat{X}} \| \mathbb{P}^{X^\star})$. The process $\widehat{X}$ follows $d\widehat{X}_t = \hat{b}_{\eta(t)}^{(N)}(\widehat{X}_{\eta(t)}) dt + \sigma_t dB_t$, while the optimal target process $X^\star$ follows $dX_t^\star = b_t^\star(X_t^\star) dt + \sigma_t dB_t$. Let $\mathcal{G}_t$ denote the filtration generated by the Brownian motion up to time $t$ and the sequence of independent random samples used to construct the drift estimators at steps $t_k \leq t$. Under the boundedness assumptions, Novikov's condition holds and Girsanov's theorem applies, so the KL divergence is given by:

$$\text{KL}(\mathbb{P}^{\widehat{X}} \| \mathbb{P}^{X^\star}) = \frac{1}{2} \int_0^1 \mathbb{E}\left[\sigma_t^{-2}\left|\hat{b}_{\eta(t)}^{(N)}(\widehat{X}_{\eta(t)}) - b_t^\star(\widehat{X}_t)\right|^2\right] dt. \tag{120}$$

Assuming $\sigma_t \geq \sigma_{\min} > 0$, we can bound the integrand by the Mean Squared Error (MSE) of the drift. We decompose the error at time $t$ into the same three components $\xi_t$, $\Delta_t$, and $D_t$ used in the Wasserstein analysis and obtain:

$$\text{KL}(\mathbb{P}^{\widehat{X}} \| \mathbb{P}^{X^\star}) \lesssim \int_0^1 \left(\mathbb{E}|\xi_t|^2 + \mathbb{E}|\Delta_t|^2 + \mathbb{E}|D_t|^2\right) dt. \tag{121}$$

We now bound the integrated MSE of each term. Crucially, notice that we must bound the integral of the expectations, which differs from the expectation of the squared integral in the Wasserstein analysis (109).

**Variance Term** ($\mathbb{E}|\xi_t|^2$). This term represents the variance of the gradient estimator. By Proposition H.4, $\mathbb{E}|\xi_t|^2 \lesssim N^{-1}$. Thus:

$$\int_0^1 \mathbb{E}|\xi_t|^2 dt \lesssim \frac{1}{N}. \tag{122}$$

**Bias Squared Term** ($\mathbb{E}|\Delta_t|^2$). By Proposition H.4, the bias satisfies $\|\Delta_t\| \lesssim N^{-1}$. Consequently, the squared bias scales quadratically:

$$\int_0^1 \mathbb{E}|\Delta_t|^2 dt \lesssim \frac{1}{N^2}. \tag{123}$$

**Discretization Term** ($\mathbb{E}|D_t|^2$). Recall that the discretization error is defined as $D_t = b_{\eta(t)}^\star(\widehat{X}_{\eta(t)}) - b_t^\star(\widehat{X}_t)$. Using the $L$-Lipschitz spatial condition and $1/2$-Hölder time condition on $b^\star$:

$$|D_t| \leq |b_{\eta(t)}^\star(\widehat{X}_{\eta(t)}) - b_t^\star(\widehat{X}_{\eta(t)})| + |b_t^\star(\widehat{X}_{\eta(t)}) - b_t^\star(\widehat{X}_t)| \lesssim \sqrt{|t - \eta(t)|} + |\widehat{X}_{\eta(t)} - \widehat{X}_t|. \tag{124}$$

Squaring and taking expectations, we apply the standard Euler-Maruyama estimate $\mathbb{E}|\widehat{X}_{\eta(t)} - \widehat{X}_t|^2 \lesssim \delta$. Since $|t - \eta(t)| \leq \delta$, we obtain the pointwise bound $\mathbb{E}|D_t|^2 \lesssim \delta$. Integrating over $[0,1]$ yields:

$$\int_0^1 \mathbb{E}|D_t|^2 dt \lesssim \delta = \frac{1}{K}. \tag{125}$$

**Completing the Bound.** Summing the contributions and keeping dominant terms yields:

$$\mathrm{KL}(\hat{p}_1 \| p_{\mathrm{reward}}) \lesssim \frac{1}{N} + \frac{1}{K}. \tag{126}$$

$\square$

## I. GLASS Flows (Holderrieth et al., 2025)

In this section, we provide additional background on GLASS Flows (Holderrieth et al., 2025). GLASS Flows provides the methodology for sampling from the conditional posterior $p_{1|t}(\cdot|x)$ of a Gaussian probability path using an ODE. The core insight is that when the prior $p_0$ is Gaussian, the drift, $\bar{b}_s$, targeting this conditional posterior can be derived by a re-parametrization of the denoiser $D_t$, and hence, the drift $b_t(x)$:

$$\bar{b}_s(\bar{x}_s; t, x) = w_1(s)\bar{x}_s + w_2(s)D_{t^*}(S(\bar{x}_s, x)) + w_3(s)x \tag{127}$$

where $w_1(s) = \frac{\dot{\alpha}_s}{\alpha_s}$, $w_2(s) = \dot{\beta}_s - \beta_s w_1(s)$, $w'_3(s) = -\bar{\gamma} w_1(s)$ are scalar coefficients, $t^*$ is a re-parametrized time, $S$ is a linear sufficient statistic, and $\bar{\gamma} = \rho\alpha_s\alpha_t$ with $-1 \leq \rho \leq 1$ denoting a free parameter (the correlation between $x_s$ and $x$ in their joint distribution conditioned on data). We can re-write this re-parametrization in terms of the unconditional drift $b_t(x)$, instead of the denoiser $D_t(x)$, as the two are related as follows:

$$D_{t^*}(S(\bar{x}_s, x)) = \frac{b_{t^*}(S(\bar{x}_s, x)) - \frac{\dot{\alpha}_s}{\alpha_s} S(\bar{x}_s, x)}{\dot{\beta}_s - \frac{\dot{\alpha}_s}{\alpha_s}\beta_s} \tag{128}$$

We further map the notations and variables in Holderrieth et al. (2025) to the notation used in our presentation of MFMs in Table 3.

| Concept | GLASS (Holderrieth et al., 2025) | MFM (Ours) |
|---|---|---|
| Clean Data | $z \sim p_{\mathrm{data}}$ | $x_1 \sim p_1$ |
| Noise | $\epsilon \sim \mathcal{N}(0, I)$ | $x_0 \sim p_0$ |
| Interpolant | $x_t = \alpha_t z + \sigma_t \epsilon$ | $I_t = \beta_t x_1 + \alpha_t x_0$ |
| Unconditional Drift | $u_t(x)$ | $b_t(x)$ |
| Denoiser | $D_t(x) = \mathbb{E}[z|x_t = x]$ | $D_t(x) = \mathbb{E}[x_1|I_t = x]$ |

*Table 3.* Translation of notation between GLASS Flows and MFM.

# J. Algorithms

---

**Algorithm 2** MFM Training (From Data)

---

**Input:** Initial parameters $\theta$ of $\hat{v}$
**while** not converged **do**
    Sample $I_0 \sim p_0$, $I_1 \sim p_1$, $t \sim \text{Unif}[0,1]$
    $I_t \leftarrow \alpha_t I_0 + \beta_t I_1$
    Sample $\bar{I}_0 \sim p_0$, $s \sim \text{Unif}[0,1]$
    $\bar{I}_s \leftarrow \alpha_s \bar{I}_0 + \beta_s I_1$
    $\frac{d}{ds}\bar{I}_s \leftarrow \dot{\alpha}_s \bar{I}_0 + \dot{\beta}_s I_1$
    # Monte Carlo loss estimates
    $\mathcal{L}_{\text{diag}} \leftarrow \|\hat{v}_{s,s}(\bar{I}_s; t, I_t) - \frac{d}{ds}\bar{I}_s\|^2$
    $\mathcal{L}_{\text{cons}} \leftarrow$ self-distillation loss (Table 2)
    $\mathcal{L}_{\text{MFM}} \leftarrow \mathcal{L}_{\text{diag}} + \mathcal{L}_{\text{cons}}$
    Compute $\nabla_\theta \mathcal{L}_{\text{MFM}}$
    Update $\theta$ by gradient descent
**end while**
**Output:** Trained parameters $\theta$

---

---

**Algorithm 3** MFM Steering (ODE)

---

**Input:** Reward $r(x)$; MFM $X$; times $0 = t_0 < \cdots < t_K = 1$; MC batch size $N$
Initialize $x_0 \sim p_0$
**for** $k = 0$ **to** $K - 1$ **do**
    $dt \leftarrow t_{k+1} - t_k$
    $\sigma_{t_k}^2 \leftarrow 2\left(\frac{\dot{\beta}_{t_k}}{\beta_{t_k}}\alpha_{t_k}^2 - \dot{\alpha}_{t_k}\alpha_{t_k}\right)$
    # Drift extraction
    $b_{t_k}(x_{t_k}) \leftarrow v_{t_k,t_k}(x_{t_k}; 0, \vec{0})$
    # Monte Carlo steering drift estimation
    Sample iid $\epsilon^{(n)} \sim p_0$ for $n = 1, \ldots, N$
    $\hat{x}_1^{(n)} \leftarrow X_{0,1}(\epsilon^{(n)}, t_k, x_{t_k})$
    Compute $\nabla\widehat{V_{t_k}(x_{t_k})}$ via (32) or (14)
    $x_{t_{k+1}} \leftarrow x_{t_k} + dt \cdot b_{t_k}(x_{t_k}) + \frac{dt \cdot \sigma_{t_k}^2}{2}\nabla\widehat{V_{t_k}(x_{t_k})}$
**end for**
**Output:** Steered sample $x_1$

---

---

**Algorithm 4** MFM Steering (SDE)

---

**Input:** Reward $r(x)$; MFM $X$; times $0 = t_0 < \cdots < t_K = 1$; MC batch size $N$
Initialize $X_0 \sim p_0$
**for** $k = 0$ **to** $K - 1$ **do**
    $dt \leftarrow t_{k+1} - t_k$
    $\sigma_{t_k}^2 \leftarrow 2\left(\frac{\dot{\beta}_{t_k}}{\beta_{t_k}}\alpha_{t_k}^2 - \dot{\alpha}_{t_k}\alpha_{t_k}\right)$
    # Drift extraction
    $b_{t_k}(X_{t_k}) \leftarrow v_{t_k, t_k}(X_{t_k}; 0, \vec{0})$
    # Score estimation by reparametrization
    Extract $\nabla \log p_{t_k}(X_{t_k})$
    # Monte Carlo steering drift estimation
    Sample iid $\epsilon^{(n)} \sim p_0$ for $n = 1, \ldots, N$
    $\hat{x}_1^{(n)} \leftarrow X_{0,1}(\epsilon^{(n)}, t_k, X_{t_k})$
    $\widehat{V_{t_k}(X_{t_k})} \leftarrow \log\left(\frac{1}{N}\sum_{n=1}^{N} e^{r(\hat{x}_1^{(n)})}\right)$
    Compute $\nabla\widehat{V_{t_k}(X_{t_k})}$ via (32) or (14)
    Sample $Z \sim \mathcal{N}(0, I)$
    $X_{t_{k+1}} \leftarrow X_{t_k} + dt\left(b_{t_k}(X_{t_k}) + \frac{\sigma_{t_k}^2}{2}\nabla\log p_{t_k}(X_{t_k}) + \sigma_{t_k}^2\nabla\widehat{V_{t_k}(X_{t_k})}\right) + \sqrt{dt}\,\sigma_{t_k}Z$
**end for**
**Output:** Steered sample $X_1$

---

---

**Algorithm 5** One-Step MFM Sampler

---

**Input:** MFM $X$
Sample $\epsilon \sim p_0$
$\hat{x}_1 \leftarrow X_{0,1}(\epsilon; 0, \vec{0})$
**Output:** Sample $\hat{x}_1$

---

---

**Algorithm 6** $K$-Step MFM Sampler

---

**Input:** MFM $X$; times $0 = t_0 < \cdots < t_K = 1$
Sample $x_0 \sim p_0$
**for** $k = 0$ **to** $K - 1$ **do**
    Sample iid $\epsilon^{(k)}, x_0^{(k)} \sim p_0$
    $\hat{x}_1^{(k)} \leftarrow X_{0,1}(\epsilon^{(k)}; t_k, x_{t_k})$
    $x_{t_{k+1}} \leftarrow \alpha_{t_{k+1}}x_0^{(k)} + \beta_{t_{k+1}}\hat{x}_1^{(k)}$
**end for**
**Output:** Sample $x_1$

---

---

**Algorithm 7** $K$-Step $\gamma = 1$ Sampler

---

**Input:** MFM $X$; times $0 = t_0 < \cdots < t_K = 1$
Sample $x_0 \sim p_0$
**for** $k = 0$ **to** $K - 1$ **do**
    Sample $x_0^{(k)} \sim p_0$
    $\hat{x}_1^{(k)} \leftarrow X_{t_k,1}(x_{t_k}; 0, \vec{0})$
    $x_{t_{k+1}} \leftarrow \alpha_{t_{k+1}}x_0^{(k)} + \beta_{t_{k+1}}\hat{x}_1^{(k)}$
**end for**
**Output:** Sample $x_1$

---

# K. Experiment Details & Additional Results

In this section, we present further discussion of our experimental settings, any hyper-parameters and additional results to supplement the main body. Across all the inference-time steering experiments, we steer the dynamics of the unconditional drift extracted from an MFM itself. Further, all baselines, namely Diffusion Posterior Sampling (DPS; (Chung et al., 2024)), Sequential Monte Carlo (SMC-TDS; Wu et al. (2024)) and Best-of-N, are all implemented using this extracted drift.

### K.1. Gaussian Mixture Model (GMM)

To analytically evaluate our proposed methods, we first consider a synthetic 2D problem.

**Model.** We consider a 2D GMM with 3 components as the prior distribution $p_1$, i.e. $p_1(x) = \sum_{i=1}^{3}\frac{1}{3}\mathcal{N}(x; \mu_i, \Sigma_i)$, where $\mu_1 = (-3, -3); \mu_2 = (0, 0); \mu_3 = (3, 3); \Sigma_1 = \Sigma_2 = \Sigma_3 = 0.5I_{2\times2}$. We train a MFM using a small MLP with the semigroup MFM loss (see 2) to sample this prior distribution.

**Reward.** We define the reward via the likelihood of a linear inverse problem with noisy measurements, $y = \boldsymbol{a}^\top\boldsymbol{x} + \epsilon$ where $\boldsymbol{a} = [1.2, -0.8]^\top$ and $\epsilon \sim \mathcal{N}(0, \sigma^2)$ with $\sigma = 0.2$. We condition on the measurement $y_{\text{obs}} = -1.0$, and as such, we target the posterior distribution $p(x|y_{\text{obs}} = -1)$ while steering.

**Sampling and Evaluation Details.** For all methods, ODEs and SDEs are solved using Euler and Euler–Maruyama schemes respectively, each with $N = 1000$ discretization steps. For SMC, we use $K = 4096$ particles and report the mean $\mathcal{S}\text{-}\mathcal{W}_2$ over 20 random seeds. For evaluation, we (i) generate 4096 posterior samples and (ii) compute sample-to-sample metrics against samples from the analytic posterior using two metrics: **(A)** sliced Wasserstein-2 (SW$_2$) and **(B)** Maximum Mean Discrepancy (MMD).

**(A)** The sliced SW$_2$ distance between distributions $P$ and $Q$ is defined as

$$\mathcal{S}\text{-}\mathcal{W}_2^2(P, Q) = \mathbb{E}_{\theta \sim \mathrm{Unif}(\mathbb{S}^{d-1})} \big[ W_2^2(\theta^\top X, \theta^\top Y) \big],$$

where $X \sim P$ and $Y \sim Q$. The expectation over $\theta$ is approximated using random one-dimensional projections.

**(B)** The squared MMD between distributions $P$ and $Q$ is defined as

$$\mathrm{MMD}^2(P, Q) = \mathbb{E}[k(X, X')] + \mathbb{E}[k(Y, Y')] - 2\,\mathbb{E}[k(X, Y)],$$

where $X, X' \sim P$ and $Y, Y' \sim Q$. We use a standard multi-scale RBF kernel and compute an unbiased empirical estimate.

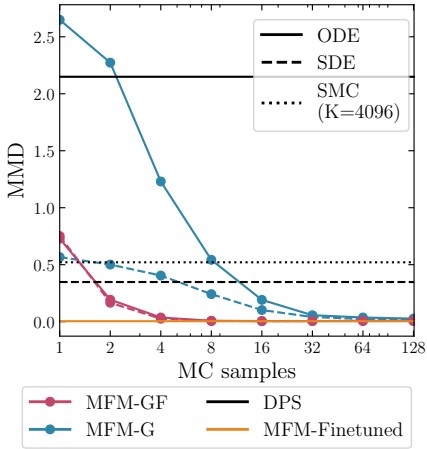

*Figure 10.* Comparison for GMM inverse problem: MMD between true posterior samples and steered samples.

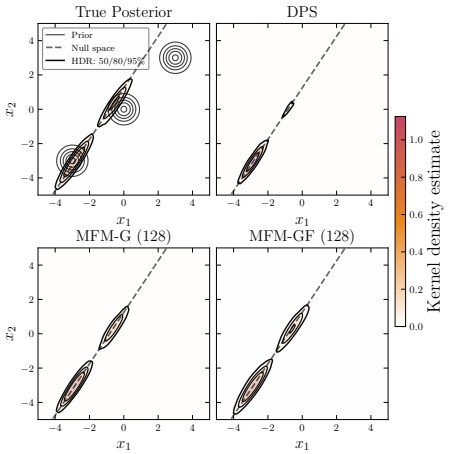

*Figure 11.* Prior, analytic posterior and density maps of steered samples (ODE).

### K.2. MNIST

**Model.** For MNIST, we train a 9M parameter UNet-based MFM using the semigroup MFM loss (see Table 2).

**Reward.** As noted in the main body, we consider a conditional sampling task with a highly-multimodal target. Specifically, we define a reward function as a weighted mixture of class probabilities obtained from a classifier: $\exp(r(x)) = p(c_{\mathrm{mix}}|x) = \sum_{i=1}^{C} w_i\, p_\theta(y_i|x)$ where $p_\theta(y_i|x)$ denotes the classifier-predicted probability that image $x$ belongs to class $i$, and the mixture weights $w_i$ satisfy $\sum_{i=1}^{C} w_i = 1$. Note that by Bayes' rule,[3] the corresponding posterior distribution takes the form $p(x|c_{\mathrm{mix}}) = \sum_{i=1}^{C} w_i\, p(x|y_i)$ which is a weighted mixture of class-conditional posteriors with weights $w_i$. By appropriately setting $\mathbf{w}$, we can define a challenging, highly multi-modal target distribution; we take $\mathbf{w} = [0, 1, 0, 1, 0, 2, 0, 2, 0, 4]$. For defining the reward model, we use a simple CNN-based classifier.

**Sampling and Evaluation Details.** As with GMMs, we use Euler and Euler-Maruyama samplers, but with $N = 500$ discretisation steps, for all methods presented. The $\mathcal{L}_2$ presented in Figure 5 is computed from the empirical class ratios observed in 4096 steered samples, and the aforementioned $\mathbf{w}$. For SMC, we use $K = 64$ particles and report the mean $\mathcal{L}_2$ over 20 random seeds. Below, we present a plot of the empirical ratios of steered samplers, for different drift estimators, alongside the ground-truth ratios, $\mathbf{w}$.

---

[3]By Bayes' rule, $p(x|c_{\mathrm{mix}}) = \sum_{i=1}^{C} p(x, y_i|c_{\mathrm{mix}}) = \sum_{i=1}^{C} p(x|y_i, c_{\mathrm{mix}})\, p(y_i|c_{\mathrm{mix}})$. If $c_{\mathrm{mix}}$ simply indexes the mixture, we identify $p(y_i|c_{\mathrm{mix}}) = w_i$ and $p(x|y_i, c_{\mathrm{mix}}) = p(x|y_i)$, yielding $p(x|c_{\mathrm{mix}}) = \sum_{i=1}^{C} w_i\, p(x|y_i)$.

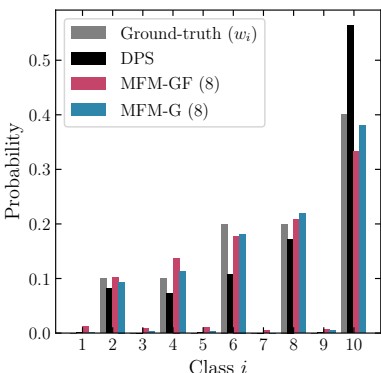

*Figure 12.* Empirical class ratios of steered samples, alongside the target ratio, **w**.

### K.3. ImageNet ($256 \times 256$)

For ImageNet ($256 \times 256$), we train MFMs by adapting DiT architectures, at B/2 and XL/2 scale, to allow for conditioning on $(t, x)$ (See G.2 for further details). These adaptations result in a small relative increase in the number of parameters, 131 $\rightarrow$ 134M and 675 $\rightarrow$ 684M respectively.

**MFM B/2.** We train a B/2 model using data. We initialise the model with the weights of a well-trained flow model, SiT B/2. As the unconditional instantaneous velocity is well learned at the start of training, the focus is on 1) learning to condition on $(t, x)$, and 2) self-distillation into a flow-map.

**MFM-XL/2.** We train XL/2 models using both data, and distillation. For both variants, we initialise at a well-trained flow-map checkpoint, DMF XL/2+ (Lee et al., 2025), for faster convergence. For the distillation variants, we regress onto the conditional drift extracted from a copy of DMF XL/2+ using GLASS Flows (see Equation 22).

**Training Objective.** When training from data, we leverage the Mean-Flow MFM objective, and for distillation, we leverage the Eulerian (Teacher) objective (See Table 2). This design choice was made to align with the standard flow-map objective used for training DMF XL/2+, the model used for initialisation. We leave a comprehensive benchmarking of the design space to future work.

**FID.** In Table 1, we present the few-step FIDs of the most performant MFM-XL/2 model, alongside deterministic flow-map baselines.

### K.3.1. ABLATIONS.

In Table 4, we present few-step FIDs of several configurations, including different model scales and objectives. In general, we took ($p_{diag} = 1.0, p_{cons} = 1.0$) as the adaptive loss coefficients (See Equation 66) following Mean Flow. However, we found that ($p_{diag} = 0.5, p_{cons} = 1.0$) was a marginally more performant configuration for 2 and 4-step generation in our XL/2 experiments. The best configuration, denoted as MFM-XL/2 in the main body and Table 1, is bolded.

*Table 4.* FID scores for increasing NFE. Lower is better. Numbers in the brackets indicate the $p$ parameter of adaptive loss for diagonal and consistency terms of the loss function, respectively. Note that $c$ parameter is fixed to be 0.01. See Table 5 for the complete set of base hyper-parameters.

| Model | NFE | | | |
|---|---|---|---|---|
| | 1 | 2 | 4 | 8 |
| **B/2** | | | | |
| Data $(1.0, 1.0)$ | 8.71 | 6.84 | 6.57 | 6.55 |
| **XL/2** | | | | |
| Data $(1.0, 1.0)$ | 4.18 | 4.18 | 4.14 | 4.41 |
| Distill $(1.0, 1.0)$ | 3.65 | 2.70 | 2.17 | 2.18 |
| **Distill $(0.5, 1.0)$** | **3.72** | **2.40** | **1.97** | **2.45** |

### K.3.2. POSTERIOR FIDELITY

We provide further details of the quantitative metrics used to evaluate the fidelity of the posterior samplers below.

**Posterior Recovery.** We first assess the fidelity of the posterior samples through FID. Concretely, we noise clean images, $\{x^{(i)}\}_{i=1}^N$ to time $t$ to form a set $\{x_t^{(i)}\}_{i=1}^N$ (where $N = 50,000$). For each noisy image, we generate a posterior sample $\hat{x}_1^{(i)} \sim p_{1|t}(\cdot \mid x_t^{(i)})$ and compute the FID between the generated set $\{\hat{x}_1^{(i)}\}_{i=1}^N$ and the set of clean images, $\{x^{(i)}\}_{i=1}^N$. We condition both the MFM and GLASS posterior samplers on the true class of $x^{(i)}$.

**Value Function Estimation.** We also evaluate the ability of the posterior samplers to support MC estimation of a value function, $V_t(x_t)$ using ImageReward (Xu et al., 2023) conditioned on the prompt *"A high-quality, high-resolution photograph of a tabby cat"* as the target reward. To establish a "ground truth" value estimate, we perform an expensive rollout of the SDE (200 steps) with $N = 200$ particles for a given noisy input $x_t$ (derived from a *tabby cat* image). We then compute cheaper MC estimates using the same particle count ($N = 200$) but via the MFM or GLASS (ODE rollout) samplers. We repeat for a set of $x_t$ obtained through noising different images of tabby cats to time $t$, and compute the correlation between the high-fidelity estimator and the cheaper estimator. We condition all three posterior samplers (for all three methods (MFM, GLASS, SDE)) on the true class *tabby cat*.

### K.3.3. INFERENCE-TIME STEERING

**Sampling and Evaluation Details.** We consider the ODE, and solve using an Euler scheme with $N = 250$ discretization steps. For MFM-G, we encountered large-magnitude steering gradients, and as such, renormalised the steering gradient (See Appendix K.3.7 for further details), which we found to be a highly performant strategy. For the Best-of-$N$ (BoN) baseline, we evaluate performance as a function of $N_{\text{BoN}} \in [1, 1000]$. To obtain a smooth curve, we first generate a pool of 128,000 samples. For each $N_{\text{BoN}}$, we partition the samples into 128 disjoint groups of size $N_{\text{BoN}}$, select the highest-scoring sample within each group according to the reward model, and report the average score across the 128 selected samples.

We present additional results on inference-time steering. For completeness, we first present the compute-normalised plot from the main body (Figure 7) with MFM-GF retained (Figure 13), noting that it performs far worse than both MFM-G and MFM-Search.

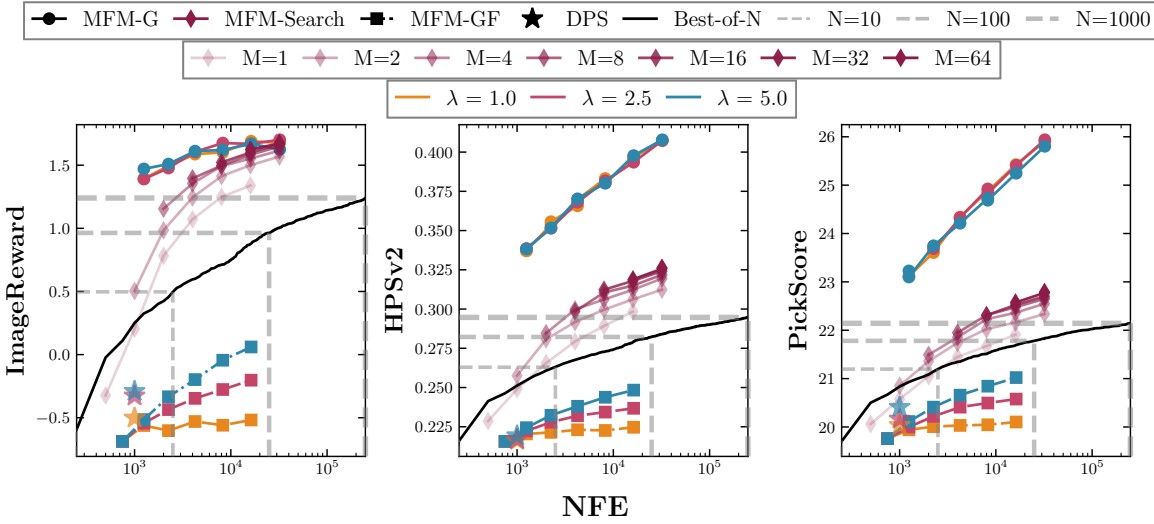

*Figure 13.* Compute-normalised performance comparison of inference-time steering schemes (with MFM-GF).

Next, we present metrics of the steered generations using reward models distinct from the one employed during steering. This serves as an important robustness check, as improvements in the steering reward should not arise from reward hacking, which would degrade performance on similar related metrics. The reward model used for steering is shown in the first (bolded) subplot, while evaluations under alternative reward models are displayed in the subsequent subplots to the right.

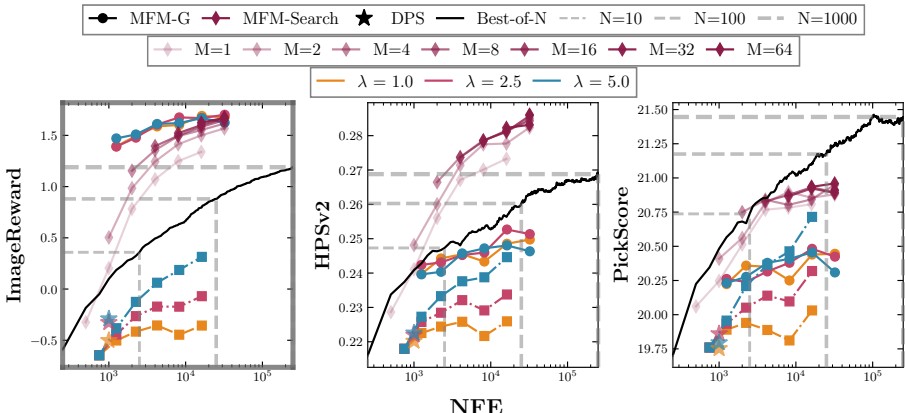

*Figure 14.* Metrics for steering using ImageReward

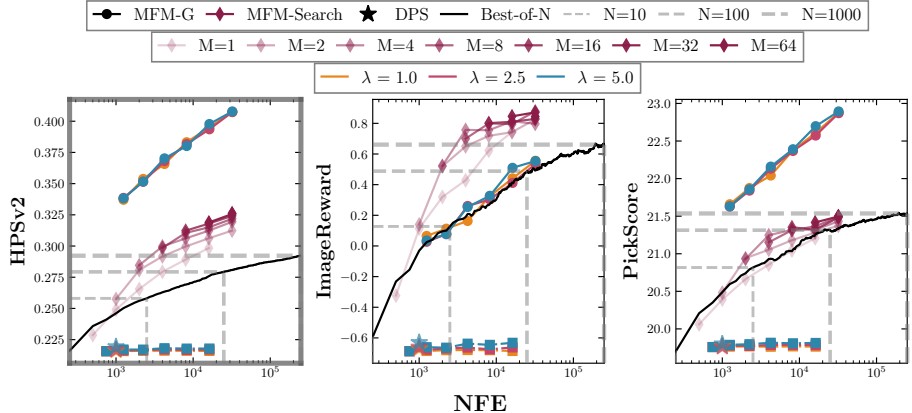

*Figure 15.* Metrics for steering using HPSv2

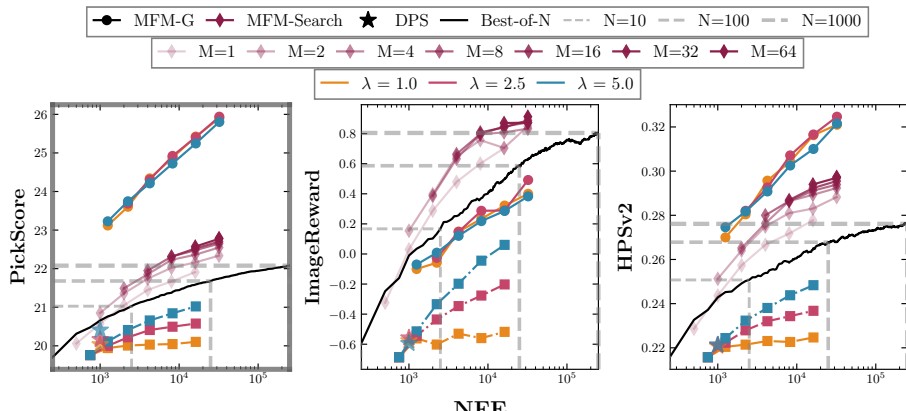

*Figure 16.* Metrics for steering using PickScore

Below, we plot the performance of MFM-GF and MFM-G against the number of MC samples in the drift estimator.

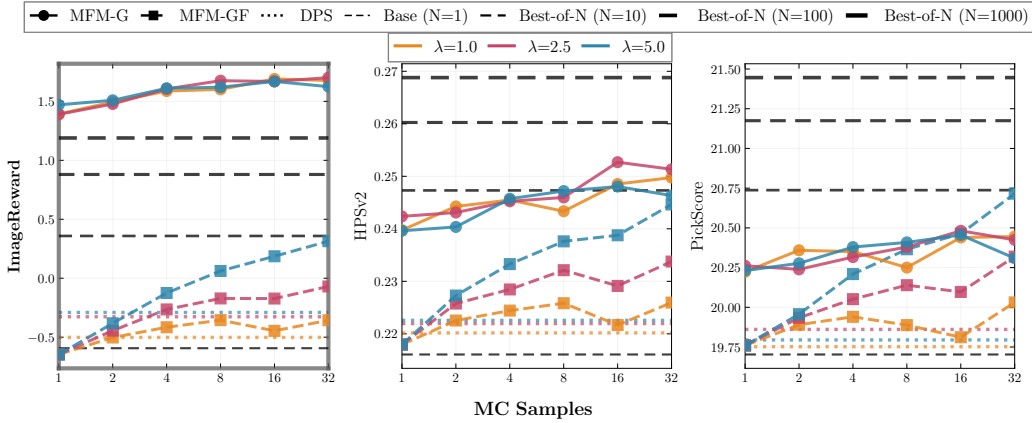

*Figure 17.* Metrics for steering using ImageReward

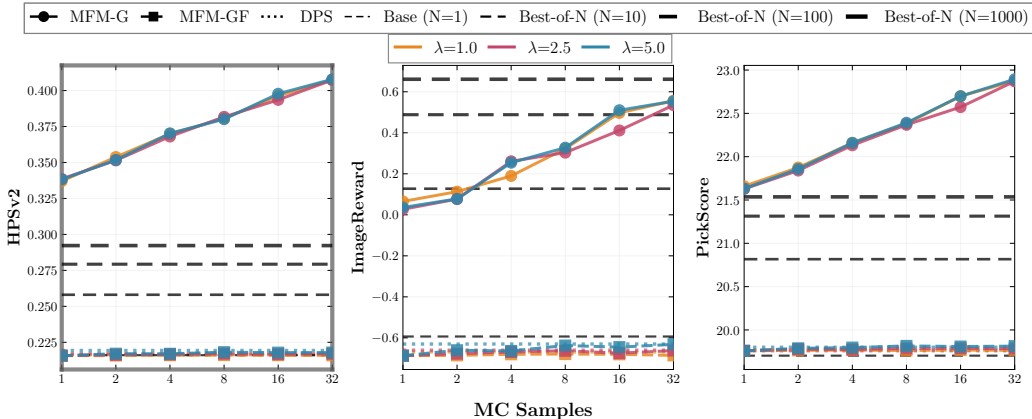

*Figure 18.* Metrics for steering using HPSv2

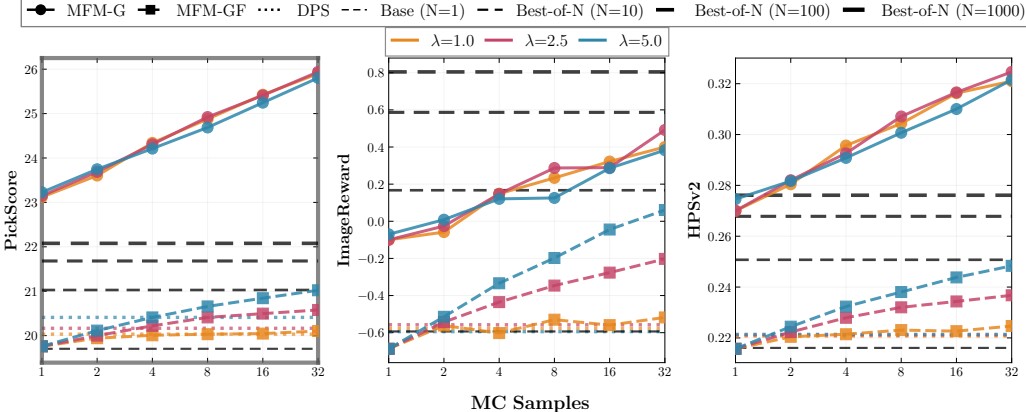

*Figure 19.* Metrics for steering using PickScore

### K.3.4. STEERED GENERATIONS

Below, we present a randomly subsampled set of generations from the Base MFM and using MFM-G for steering (HPSv2 reward model), for $\{1, 2, 4, 8, 16, 32\}$ samples in the MC estimate. Note that in each column, each row of generations share the same random seed.

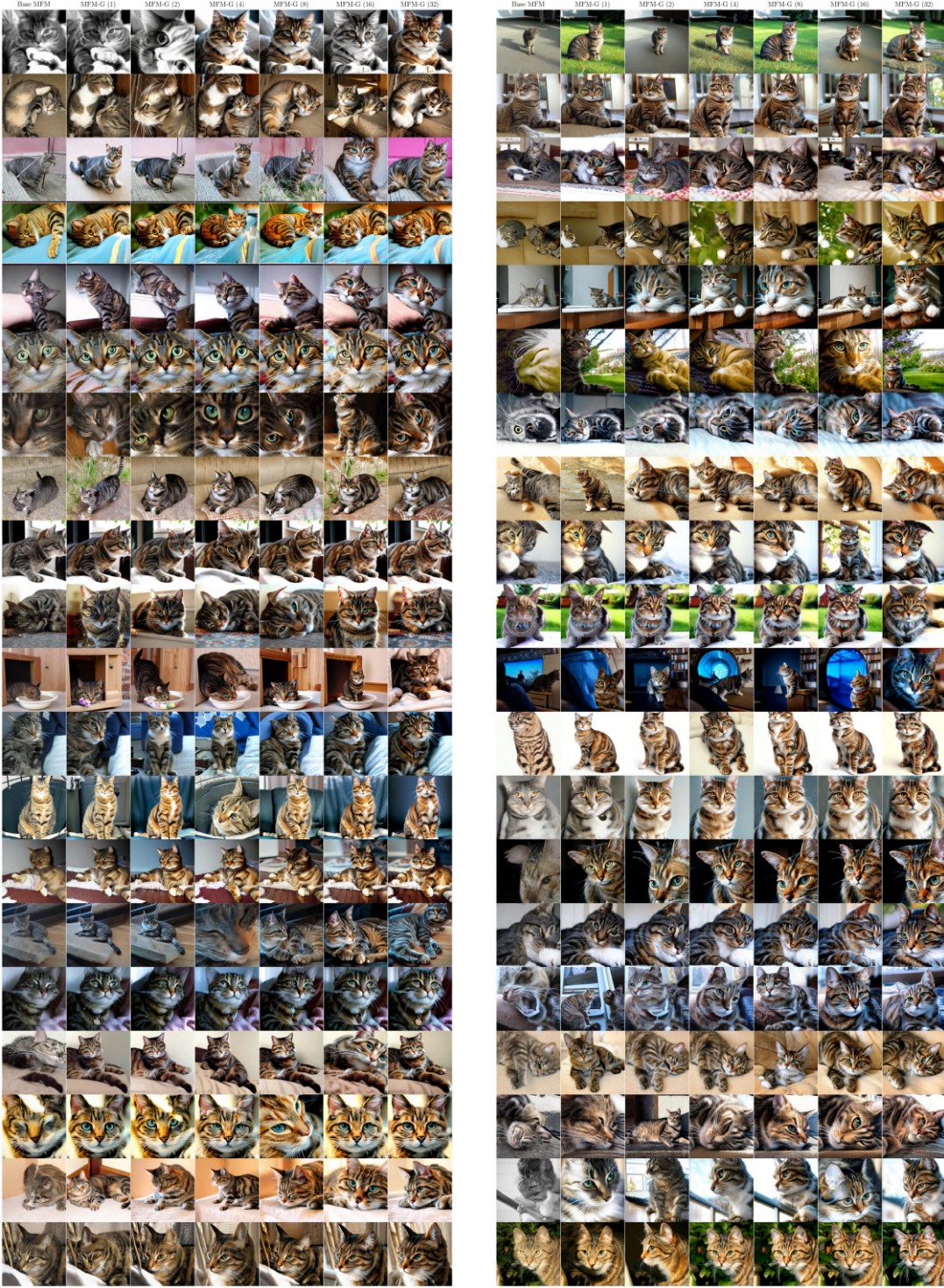

*Figure 20.* Base MFM and MFM-G steered generations for 30 random seeds (HPSv2).

## K.3.5. FINE-TUNED GENERATIONS

Below, we present a randomly subsampled set of generations from the base and MFMs finetuned using $\lambda = \{10, 25, 50\}$.

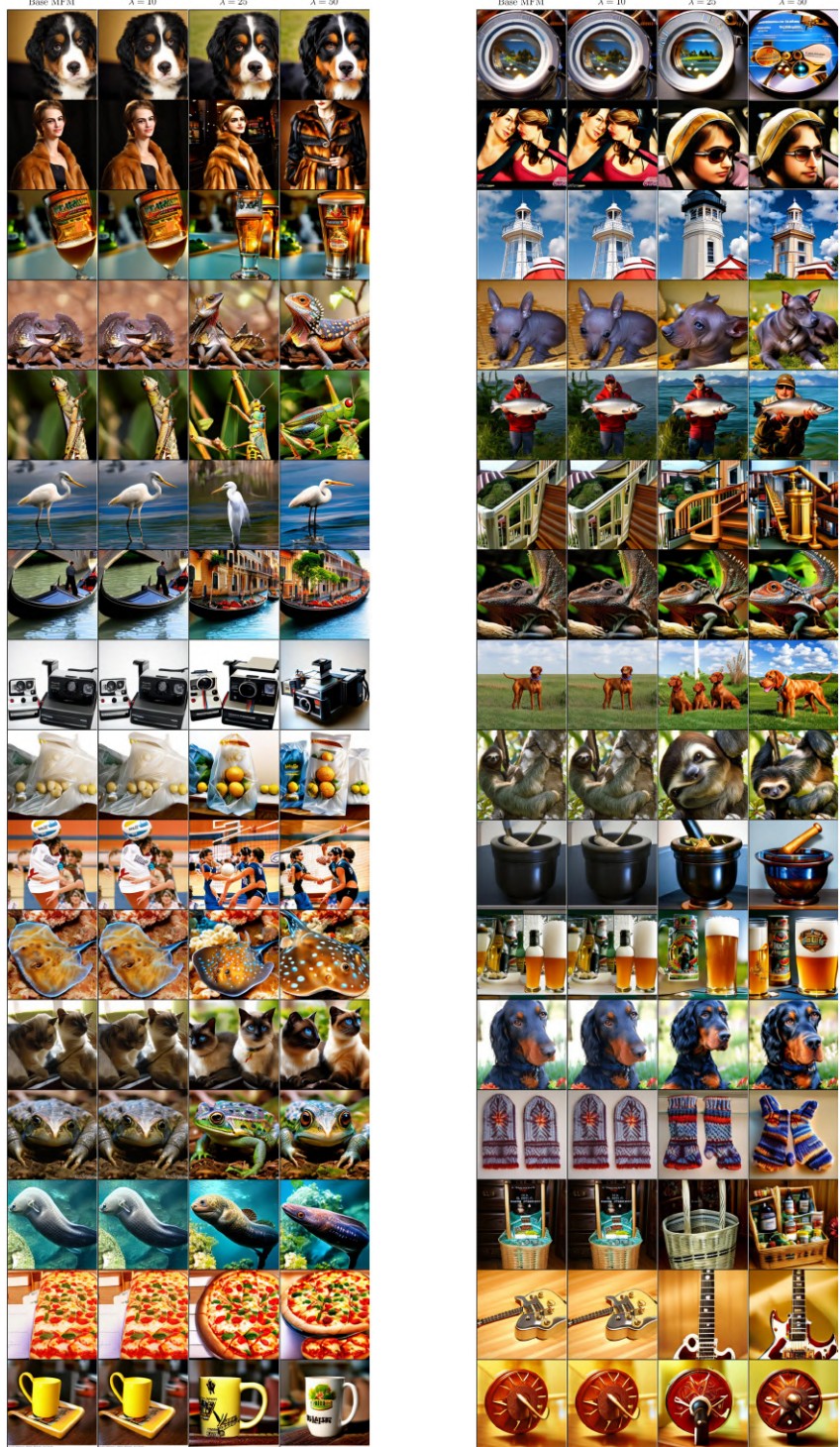

*Figure 21.* Base MFM and fine-tuned MFM generations for 32 random seeds (HPSv2).

### K.3.6. NUMBER OF FUNCTION EVALUATIONS (NFES)

**MFM-GF.** For a $K$-step discretisation of the continuous-time ODE, with $N$ posterior samples for drift estimation at each step, MFM-GF requires

$$\text{NFE} = K + 2NK.$$

This consists of $K$ evaluations of the base drift, and at each step $N$ one-step posterior samples and $N$ reward evaluations to estimate the value function.

**MFM-G.** For a $K$-step discretisation of the continuous-time ODE, with $N$ posterior samples for drift estimation at each step, MFM-G requires

$$\text{NFE} = K + 4NK.$$

This consists of $K$ base drift evaluations and, at each step, $N$ one-step posterior samples and $N$ reward evaluations for value estimation. MFM-G additionally requires a backward pass through the $2N$ network evaluations at each step; which we assume incurs a $2\times$ multiplicative cost.

**MFM-Search.** For a $K$-step discretisation, with $M$ candidate solutions and $N$ posterior samples for drift estimation at each step, MFM-Search requires

$$\text{NFE} = 2MNK$$

This consists of $M * N$ one-step posterior samples, and $M * N$ reward evaluations (i.e. one per posterior sample) at each step.

**DPS.** For a $K$-step discretisation of the continuous-time ODE, DPS requires

$$\text{NFE} = 4NK.$$

At each step, DPS requires a base drift evaluation, from which the Tweedie estimate can be recovered, and a reward function evaluation. As with MFM-G, we assume a $2\times$ multiplicative cost for the backward pass used to compute the gradient.

**Best-of-$N$ baseline.** Using $N_{\text{BoN}}$ samples, the Best-of-$N$ baseline requires

$$\text{NFE} = KN_{\text{BoN}} + N_{\text{BoN}}.$$

This corresponds to generating $N_{\text{BoN}}$ samples using $K$ discretisation steps, followed by a final reward evaluation for each sample required for selecting the highest-reward sample.

### K.3.7. NORMALISATION OF STEERED DRIFT ESTIMATOR

In ImageNet experiments, we encountered steering drifts much larger in magnitude than the unconditional drift when using the MFM-G drift estimator. In order to faithfully realise the steered dynamics in such settings, we would require a much finer time discretisation than is required for unconditional generation to avoid excessive discretisation error. For practical implementation, we considered two different solutions, 1) *clipping* and 2) *rescaling* the steering drift relative to the norm of the unconditional drift. Although this introduces a bias, it is only in the magnitude of the drift and not the direction. As it was both highly stable and performant, we used the *rescaling* in all our ImageNet inference-time steering experiments, with $\lambda = 1$:

$$b_t^*(x) = b_t(x) + \lambda \left\| b_t(x) \right\|_2 \frac{\nabla V(x)}{\left\| \nabla V(x) \right\|_2}. \tag{129}$$

### K.3.8. IMAGENET TRAINING CONFIGURATION

We present below the base hyper-parameters used in training the models presented in Table 4.

*Table 5.* Configurations for ImageNet experiments.

|  | MFM-B/2 | MFM-XL/2 |
|---|---|---|
| *Model* | | |
| Resolution | 256×256 | 256×256 |
| Params (M) | 134 | 683 |
| Hidden dim. | 768 | 1152 |
| Heads | 12 | 16 |
| Patch size | 2×2 | 2×2 |
| Sequence length | 256 | 256 |
| Layers | 12 | 28 |
| Encoder depth | 8 | 20 |
| *Optimization* | | |
| Optimizer | AdamW (Kingma & Ba, 2017) | |
| Batch size | 256 | |
| Learning rate | 1e-4 | |
| Adam $(\beta_1, \beta_2)$ | (0.9, 0.95) | |
| Adam $\epsilon$ | 1e-8 | |
| Adam weight decay | 0.0 | |
| EMA decay rate | 0.9999 | |
| *Flow model training* | | |
| Training iteration | 800K | 4M |
| Epochs | 160 | 800 |
| Class dropout probability | 0.2 | 0.2 |
| Time proposal $\mu_{\text{FM}}$ | 0.0 | - |
| REPA alignment depth | - | 8 |
| REPA vision encoder | - | DINOv2-B/14 |
| QK-norm | 55 | 55 |
| *DMF flow map training* | | |
| Training iteration | - | 400K |
| Epochs | - | 80 |
| Class dropout probability | - | 0.1 |
| Time proposal $\mu_{\text{FM}}$ | - | 0.0 |
| Time proposal $(\mu_{\text{MF}}^{(1)}, \mu_{\text{MF}}^{(2)})$ | - | (0.4, -1.2) |
| Model guidance scale $\omega$ | - | 0.6 |
| Guidance interval | - | [0.0, 0.7] |
| *MFM training* | | |
| Training iteration | 100K | 100K |
| Batch size | 512 | 360 |
| Epochs | 40 | 28 |
| Optimizer | RAdam (Liu et al., 2021) | |
| Learning rate | 1e-4 | |
| Learning rate warmup | Linear (first 2000 steps) | |
| RAdam $(\beta_1, \beta_2)$ | (0.9, 0.999) | |
| RAdam $\epsilon$ | 1e-8 | |
| RAdam weight decay | 0.0 | |
| EMA decay rate | 0.9999 | |
| Class dropout probability | 0.2 | |
| Model guidance scale $\omega$ | 0.6 | |
| Guidance interval | [0.0, 1.0] | |

