# OpenReview forum: "Meta Flow Maps enable scalable reward alignment"
_ICML.cc/2026/Conference — ICML 2026 regular_

### Official Review · Reviewer_BUSz · 2026-03-05

**Soundness:** 3
**Presentation:** 3
**Significance:** 4
**Originality:** 4
**Overall Recommendation:** 5
**Confidence:** 3

**Summary:**

The paper proposes Meta Flow Maps (MFMs), a novel extension of flow map models that amortizes an infinite family of conditional vector fields into a single neural operator. Instead of learning a single transport from prior to data, the model learns a context-conditioned flow map capable of generating one-step samples from the conditional posterior. This effectively turns a trained flow model into a posterior sampler without requiring expensive trajectory rollouts.

In addition to being competitive as a few-step generative model for unconditional sampling, MFMs enable efficient posterior sampling at arbitrary intermediate states. The authors demonstrate the practical value of this capability in reward-aligned generation, where posterior sampling is used to estimate value function gradients for inference-time steering and off-policy fine-tuning.

**Compliance With Llm Reviewing Policy:**

Affirmed.

**Final Justification:**

The rebuttal confirmed my initial assessment of the paper.

**Key Questions For Authors:**

- The value-function estimator relies critically on the quality of posterior samples. Could the authors provide further analysis or ablations quantifying how sensitive steering performance is to posterior approximation error, such as under imperfect MFM training or low-capacity settings?
- If I understand correctly, inference-time steering still requires integrating the controlled ODE and estimating the value-function gradient at each timestep, whereas fine-tuning distills this correction into the drift and removes the need for Monte Carlo posterior sampling during inference. Could the authors elaborate on the practical trade-offs between these two approaches? In particular, how should one decide between steering and fine-tuning in terms of computational cost, flexibility across rewards, and performance?
- While MFMs eliminate inner posterior rollouts, inference-time steering still requires integrating the controlled ODE. Do the authors see a pathway toward distilling the steered dynamics into a single-step generator?

**Limitations:**

The paper does not explicitly discuss limitations. While MFMs are competitive as few-step generative models, they do not consistently outperform state-of-the-art deterministic few-step baselines and require additional training on top of a pretrained flow model. Moreover, although MFMs eliminate expensive inner posterior rollouts, inference-time steering still requires integrating the controlled ODE over time. Practical considerations such as Monte Carlo variance in value-function gradient estimation and potential degradation in posterior approximation quality are not extensively analyzed. A clearer discussion of these aspects would strengthen the paper.

**Strengths And Weaknesses:**

- Soundness: The paper is technically sound and carefully developed. The proposed Meta Flow Map (MFM) framework is well grounded in the theory of flow matching and stochastic interpolants, and the derivations linking posterior sampling to value-function gradient estimation are convincing. The training objectives are well motivated, and the empirical results support the main claims. In particular, the method clearly delivers on its core objective, amortizing conditional vector fields to enable efficient posterior sampling from a pretrained flow model. Experimental validation across synthetic settings and ImageNet further strengthens the soundness of the approach.

- Presentation: The paper is clearly written, well structured, and generally easy to follow despite the technical depth. The conceptual transition from standard flow maps to Meta Flow Maps is well articulated, and the empirical sections are thorough. The narrative consistently ties the method to the central bottleneck of posterior sampling for reward alignment.

- Significance: The paper addresses the problem of scalable reward alignment and posterior sampling in transport-based generative models. By amortizing an infinite family of conditional ODEs, the method meaningfully advances the computational efficiency of inference-time steering and off-policy fine-tuning.

- Originality: The introduction of Meta Flow Maps as an operator that amortizes a family of conditional vector fields is conceptually novel. The work provides a well-motivated extension of flow maps, turning them into posterior samplers rather than only marginal generators.

---

> ### Author Rebuttal · Authors · 2026-03-30
>
> We thank the reviewer for their kind review.
>
> ### Answer to Question 1:
>
> Thank you for this suggestion. We agree that the value function estimator depends on posterior sample quality, so we quantify this directly in the GMM setting, where both the exact conditional posterior and the exact steered target are available. We evaluate intermediate MFM checkpoints throughout training and measure (i) posterior approximation error via RBF MMD between one-step posterior samples and the true conditional posterior, and (ii) steering error via RBF MMD between steered samples and the exact steered target. The results show a clear monotone relationship: under-trained checkpoints with worse posterior approximation error also yield worse steering, and as posterior quality improves, steering performance improves. We include the figure at the **following link**: https://drive.google.com/file/d/1QZ-1FjN8hun8WBgUIUpciF-1MdJ6jfJv/view?usp=sharing. This provides a direct ablation showing how posterior approximation quality translates into downstream steering performance under imperfect MFM training.
>
> ### Answer to Question 2:
>
> Your understanding is completely correct. Steering and fine-tuning target the same controlled drift, but they pay the cost in different places. Steering has no upfront cost for a new reward, but generation is more expensive because the reward correction (value function gradient) must be estimated along the trajectory at inference time. Fine-tuning is the opposite, as it has a larger upfront cost, but once trained, inference is much cheaper because the correction has been amortized into the model.
>
> In practice, steering is preferable when the reward changes often or when one wants to try many rewards without retraining. Fine-tuning is preferable when the reward is fixed and the aligned model will be used many times, so the upfront cost can be amortized.
>
> In terms of performance, an important factor is how far the reward-tilted target is from the base model distribution. If the target is sufficiently out of distribution for the base model, fine-tuning is often preferable. In that regime, the base drift $b_t(x)$ may simply not be well trained in the regions that matter under the new reward. Fine-tuning lets the model adapt its drift $\hat b_t^{\star}(x)$ to those important regions directly. At the same time, steering remains attractive for complex reward functions, since keeping the reward in the loop at inference time can help. More broadly, we view the two approaches as complementary, and in some settings they can also be combined.
>
> ### Answer to Question 3:
>
> Yes, this is definitely a very natural next step. MFMs remove the need for inner posterior rollouts when estimating the drift correction during inference-time steering. MFM fine-tuning goes a step further by distilling this controlled drift directly into a new model, eliminating the need to estimate it during inference. A natural next step is then to compress the resulting controlled ODE into a few-step sampler, for example a flow map or MFM targeting the reward-tilted distribution $p_{\text{reward}}$. This can be done either by self-distillation jointly during fine-tuning, or in two stages by first fine-tuning and then distilling afterward. Either route produces a one or few-step generator for $p_{\text{reward}}$. We thank the reviewer for raising this point and will make this connection more explicit in the revised paper.
>
> ### Regarding Limitations:
> We thank the reviewer for the thoughtful list of points that require further discussion. We agree, and will aim to flesh these out in further depth for the camera-ready version if accepted. In particular, we will move the discussion from Appendix H.2 (Formal Convergence Guarantees) to the main body, which discusses the effects of Monte Carlo variance.

---

> > ### Author Rebuttal · Reviewer_BUSz · 2026-04-02
> >
> > I thank the authors for carefully answering my questions, for the clarifications and for the added results. I will keep my score.

---

> > > ### Author Response · Authors · 2026-04-06
> > >
> > > We are pleased to hear that our rebuttal has addressed your questions, and we would like to thank the reviewer for their positive and helpful comments during the review process.
> > >
> > > Thanks,
> > >
> > > The Authors

---

### Official Review · Reviewer_mGKK · 2026-03-12

**Soundness:** 3
**Presentation:** 4
**Significance:** 3
**Originality:** 3
**Overall Recommendation:** 5
**Confidence:** 3

**Summary:**

This work proposes a stochastic extension of consistency models that can directly sample from reward-tilted conditional posterior distributions from intermediate states, and it presents a unified view of inference-time steering and training-time fine-tuning through value gradient estimation. Building on this framework, the paper further introduces MFM-FT, an unbiased and off-policy objective for fine-tuning, and demonstrates its effectiveness from toy problems to ImageNet.

**Compliance With Llm Reviewing Policy:**

Affirmed.

**Final Justification:**

The author's rebuttal has addressed my concerns. I will leave the rating as “accepted”.

**Key Questions For Authors:**

I have decided to accept this paper based on its content, but I would appreciate it if the authors could further address the above weakness, especially by discussing the trade-off between training cost and inference efficiency.

**Limitations:**

Yes

**Strengths And Weaknesses:**

Strengths
- The paper is well written and very easy to follow.
- The proposed method is a natural stochastic extension of consistency models, and is based on the simple and intuitive idea of directly generating samples from the conditional posterior distribution from intermediate states. At the same time, it provides a unified view of inference-time steering and training-time fine-tuning through the common framework of value gradient estimation, and gives a theoretical foundation based on stochastic optimal control, which strengthens the validity of the method. In particular, the introduction of MFM-FT, an unbiased and off-policy objective for fine-tuning, is an important technical contribution of this work.
- The experiments cover a wide range of settings, from toy data to high-resolution ImageNet, and demonstrate effectiveness for both inference-time steering and fine-tuning. Moreover, the ImageNet experiments are evaluated with text prompts and multiple general-purpose reward models, and it is also a strength that the paper attempts to verify effectiveness across diverse settings.

Weakness
- A main strength of the method is the reduction in inference cost through few-step inference. However, the training cost is not small. On ImageNet, in addition to training the base model (800 epochs), the method requires training a DMF flow map (80 epochs), followed by additional training of the MFM (about 30 epochs). The best performance also depends on a strong pretrained flow map and distillation. Therefore, to more convincingly demonstrate the benefit of reduced inference cost, it would be helpful to include further discussion of the trade-off with training cost, as well as the kinds of use cases in which this upfront cost is justified.

---

> ### Author Rebuttal · Authors · 2026-03-30
>
> We thank the reviewer for their kind review.
>
> We now address their important point regarding the trade-off between the additional training cost and the efficiency gains that MFMs provide.
>
> Our main intended use case is large generative models that are pre-trained once and then reused across many downstream tasks. In this setting, we believe there is substantial value in equipping a pretrained model with MFM capabilities as part of the overall training pipeline. The additional training required to convert such a model into an MFM is relatively small compared to the total cost of training the underlying model, while significantly increasing its utility for downstream control.
>
> More broadly, our goal is to make reward alignment a more intrinsic capability of the generative model, rather than something that must be solved again from scratch for each new objective. In other words, we want pretrained models to be more readily adaptable to new rewards, constraints, and steering objectives after pretraining, without requiring expensive task-specific procedures each time.
>
> This is particularly valuable in settings where the same large model must be used for many different tasks. Examples include image or video generation, where users may wish to use in-the-loop reward functions at inference time, and protein design, where a large generative model may be queried repeatedly under many different reward functions or design constraints. In such scenarios, reward-specific fine-tuning is often impractical, and repeated steering can itself become a major computational burden. MFMs help address this by making both steering and downstream fine-tuning significantly more computationally tractable.
>
> We will revise the paper to emphasize this motivation more clearly, and to better discuss the settings in which this additional upfront training cost is justified by repeated downstream use.

---

> > ### Author Rebuttal · Reviewer_mGKK · 2026-04-02
> >
> > Thank authors for the response. My concerns are addressed. I will keep my score.

---

> > > ### Author Response · Authors · 2026-04-06
> > >
> > > We are pleased to hear that our rebuttal has addressed your question, and we would like to thank the reviewer for their suggestions.
> > >
> > > Best,
> > >
> > > The Authors

---

### Official Review · Reviewer_yLSG · 2026-03-12

**Soundness:** 4
**Presentation:** 3
**Significance:** 4
**Originality:** 3
**Overall Recommendation:** 4
**Confidence:** 5

**Summary:**

Meta Flow Maps (MFM) introduces a framework for making reward alignment in diffusion and flow-based generative models more scalable. The main contribution is a conditional model that, given an intermediate noisy state xt, can produce stochastic one-step samples of the clean endpoint distribution p(x1 | xt) in a single call, while remaining differentiable with respect to xt. This enables efficient Monte Carlo estimation of value functions and value gradients without requiring costly multi-step reverse-time rollouts.

The paper presents training procedures to learn these conditional flow maps from data using coupled stochastic interpolants, and also describes a teacher-distillation route that uses an analytic conditional-drift construction to improve training efficiency. Building on the learned one-step posterior sampler, the authors derive methods for inference-time steering and for off-policy reward fine-tuning that leverage the amortized posterior sampling capability. Experiments on ImageNet-scale generation and multiple reward models show that the approach can maintain competitive few-step generation quality while improving compute-normalized performance for reward-driven steering and supporting efficient reward-oriented fine-tuning.

**Compliance With Llm Reviewing Policy:**

Affirmed.

**Key Questions For Authors:**

Posterior sampler validity
For a fixed xt, do one-step samples match the intended conditional endpoint distribution and behave like independent draws? Please share direct diagnostics (e.g., conditional distance to a teacher conditional, diversity/coverage, simple correlation/ESS). Strong evidence would increase my confidence; signs of mode collapse or strong dependence would reduce it.

Steering robustness
How sensitive are results to the number of posterior samples per xt and key guidance hyperparameters? If performance is stable without heavy per-reward tuning, I would view the method as more practically significant; brittleness would lower my assessment.

Fine-tuning trade-offs
Does the off-policy objective improve reward without harming diversity/coverage or inducing reward hacking, compared to common heuristic posterior approximations? If quality and diversity are preserved, my evaluation improves; if not, it weakens.

Fair compute accounting
Can you report model NFEs and reward-model evaluations separately and match baselines under the same total budget? If advantages persist under strict accounting, it strengthens the claims; if not, it weakens them.

**Limitations:**

yes

**Strengths And Weaknesses:**

Strengths

Soundness
The paper is technically well motivated and targets a real bottleneck in diffusion/flow alignment: estimating conditional expectations and value gradients typically requires expensive multi-step reverse-time simulation. Learning a stochastic one-step conditional sampler with reparameterized noise is a coherent way to enable differentiable Monte Carlo value-gradient estimates. The two training routes are plausible: data-driven coupled interpolants provide principled conditional supervision, and the analytic teacher-distillation construction is a reasonable variance-reduction mechanism. Experiments cover both base generation quality (few-step ImageNet) and downstream alignment (steering under multiple rewards and reward fine-tuning), which is the right structure to support the claims.

Presentation
The paper is generally well organized with a clear progression from motivation to method, training, and applications. The main idea is communicated at a high level, and the connection to steering and fine-tuning is clear.

Significance
If reliable, amortized one-step conditional sampling can substantially reduce the compute cost of reward alignment and make steering/fine-tuning more practical. The emphasis on compute-normalized comparisons is relevant for real-world use.

Originality
The contribution is a meaningful reframing and integration: turning alignment-time rollouts into a learned stochastic one-step conditional sampler and building steering and off-policy fine-tuning around it. Even if components relate to prior flow/consistency and guidance ideas, the “meta” conditional sampling capability and the analytic distillation route are a nontrivial combination.

Weaknesses

Soundness
The strongest claims depend on how well the model matches the intended conditional endpoint distribution for a fixed intermediate state and whether samples behave like i.i.d. draws. Evidence is mostly indirect (reward and FID), so direct conditional-distribution diagnostics would strengthen the “posterior sampler” claim. The off-policy “unbiased” fine-tuning objective is appealing but would benefit from stronger empirical checks against common failure modes (diversity loss, mode collapse, reward hacking) and comparisons to simpler posterior approximations.

Presentation
Key derivations are dense and can be hard to follow. Reproducibility would improve with a single algorithm box, explicit training/sampling recipes, and clearer compute accounting that separates model NFEs from reward-model evaluations, especially versus Best-of-N baselines.

Significance
It is not fully clear how broadly the benefits transfer across tasks, modalities, or reward types, or in which regimes the extra training complexity is most worthwhile compared to simpler guidance methods.

Originality
Novelty is primarily in the reframing and system-level integration rather than a completely new primitive. The paper could more explicitly differentiate from closely related stochastic/conditional consistency or distillation variants.

---

> ### Author Rebuttal · Authors · 2026-03-30
>
> We thank the reviewer for their kind review. Please see additional figures and tables at the **following link**: https://drive.google.com/file/d/1HNYpBFRtWTtIZGcR6-YjYBH0uZHVNFtd/view?usp=sharing.
>
> ### Answer to question 1:
>
> We agree that direct conditional distribution diagnostics are important. In the paper, we already provide three complementary pieces of evidence: (1) qualitative conditional endpoint samples for fixed $x_t$, which show diverse one-step draws that become appropriately more concentrated as the conditioning time increases (Figure 4); (2) posterior recovery on ImageNet 256 (Figure 12); and (3) value function estimation accuracy using MFM posterior samples, compared against a high-fidelity reference sampler (Figure 12). Together, these results already suggest that the learned one-step sampler captures the intended conditional law well.
>
> To further address the reviewer's question, we additionally ran two new diagnostics on ImageNet 256 against a high-fidelity SDE posterior sampler. First, in the left subfigure of Figure 1 from the link, we compare the variance of one-step MFM samples and SDE samples across conditioning times in Inception feature space, which is widely used for distributional evaluation of images (e.g. FID). The two variance curves track each other closely over the full range of $t$, which indicates that the learned sampler preserves posterior spread and does not exhibit obvious variance collapse. Second, in the right subfigure, we report CRPS, which is a standard distributional metric, again in inception feature space. Here again, MFM closely matches the SDE reference across conditioning times, providing direct evidence that the learned one-step sampler remains close to the target conditional distribution for a given $x_t$.
>
> Overall, we view the original posterior-recovery and value-estimation results, together with these new variance and CRPS diagnostics, as supporting that MFM one-step samples are both diverse and well matched to the intended conditional endpoint distribution.
>
>
> ### Answer to question 2:
>
> The only user specified quantity is the tilt strength $\lambda$, which determines the target distribution $p_1(x) \exp (\lambda r(x))$. Once $\lambda$ is fixed, we do not tune or introduce any guidance hyperparameters.
>
> Regarding the number of posterior samples, Figure 6 reports inference-time steering results for $M \in$ {$1,2,4,8,16,32$} where $M$ is the number of posterior samples per $x_t$. As expected, performance improves monotonically with $M$, since the Monte Carlo estimate of the steering drift becomes more accurate and lower variance as more posterior samples are used. At the same time, the method is already effective in the low sample regime. Even with a single posterior sample ($M=1$), MFM-G outperforms all baselines, showing that steering performance is robust to the number of posterior samples.
>
>
> ### Answer to question 3:
>
> Yes, MFM-FT avoids the bias introduced by common heuristic posterior approximations. Specifically, Eq. (25) defines an unbiased objective whose unique fixed point is the optimal reward tilted drift $b_t^\star$. Therefore, the method directly targets the reward tilted distribution $p_1(x)\exp(r(x))$.
>
> Empirically, we evaluate potential reward hacking by fine-tuning the MFM using only HPSv2, while reporting performance on two additional reward metrics in Figure 8. We observe that HPSv2 improves steadily throughout training, and that PickScore and ImageReward improve as well. This suggests that the gains are not simply due to reward hacking or overfitting to a single reward model, but instead reflect broader improvements in sample quality. In addition, Figure 3 shows that the fine-tuned MFM produces visually higher-quality samples while preserving the semantic content of the corresponding base samples, providing qualitative evidence that alignment is improved without obvious semantic collapse.
>
> ### Answer to question 4:
>
> Our current accounting in terms of total NFEs follows the standard convention in the inference-time steering literature, but we agree that an explicit decomposition is also helpful. For this reason, we provided Appendix K.3.6, which specified the exact NFE accounting for each steering method. We thank the reviewer for their suggestion, and will present this information as a table with base model forward calls, base model backward passes, reward model forward calls, and reward model backward passes for every method. The resulting breakdown is shown in Table 1 of the link.
>
> For the closest like-for-like comparison, we compare MFM-G against DPS under this stricter decomposition, since both methods use reward gradients during sampling. The matched-budget comparison is shown in Table 2 of the link. Under this stricter accounting, MFM-G still substantially outperforms DPS.
>
> We hope that our responses address the reviewer’s main concerns, and we kindly ask them to consider increasing their score if they agree.

---

> > ### Author Rebuttal · Reviewer_yLSG · 2026-04-03
> >
> > Thanks authors for submitting rebuttal.  I am keeping my score

---

> > > ### Author Response · Authors · 2026-04-03
> > >
> > > We thank the reviewer for the positive assessment and for confirming that their concerns have been addressed.
> > >
> > > If there are any remaining issues or considerations that are keeping the score at its current level, we would greatly appreciate any additional feedback.

---

### Official Review · Reviewer_abph · 2026-03-18

**Soundness:** 4
**Presentation:** 4
**Significance:** 3
**Originality:** 3
**Overall Recommendation:** 4
**Confidence:** 4

**Summary:**

This paper proposes Meta Flow Maps, a stochastic extension of flow maps that learns to perform one-step conditional posterior sampling $p_{1|t}(\cdot|x)$ for any intermediate state $(t,x)$ in a flow-based generative model. Experiments on GMM, MNIST, and ImageNet 256 demonstrate competitive unconditional generation and strong steering performance.

**Compliance With Llm Reviewing Policy:**

Affirmed.

**Final Justification:**

I thank the authors for their responses. I maintain my current score.

**Key Questions For Authors:**

1. How does MFM posterior quality degrade as dimensionality increases beyond ImageNet 256?

2. How does MFM-FT compare to REFL or GRPO under similar compute budgets? It's also fine not to run experiments — I'd like to see more analysis, or perhaps the method you propose is actually compatible with theirs.

**Limitations:**

The experimental evaluation is narrow — a single dataset/resolution with no comparison to the most relevant RL fine-tuning baselines.

**Strengths And Weaknesses:**

## Strengths

- The formulation is clean. Reducing stochastic flow map learning to amortized flow map learning over a family of conditional ODEs is elegant and well-motivated.
- Unifying steering and fine-tuning under a single framework (value function gradient estimation) is theoretically satisfying, and the unbiased off-policy fine-tuning objective is a nice contribution.
- The paper is generally well-written with clear exposition of the mathematical framework.

## Weaknesses

- Limited experimental scope. Only ImageNet 256 class-conditional, with a fixed prompt template for T2I rewards. No comparison with recent RL-based alignment methods (GRPO, REBEL, etc.) that are the natural competitors for fine-tuning. The GMM and MNIST experiments are toy-scale.

---

> ### Author Rebuttal · Authors · 2026-03-31
>
> We thank the reviewer for their kind review and for finding our formulation elegant.
>
>
> ### Answer to Question 1:
>
> We expect MFM posterior quality to scale favourably with dimensionality, since MFMs build directly on flow matching and flow map training, both of which have been shown to perform well in high-dimensional settings. That said, we agree that the paper would benefit from additional empirical evidence on scaling. As such, we trained an MFM on ImageNet 512 ($4\times64\times64=16,384$ latent dim), compared to ImageNet 256 ($4\times32\times32=4,096$ latent dim). We used the same hyperparameters as in our best ImageNet 256 run; further gains are likely possible with additional training and hyperparameter tuning.
>
> We present below the unconditional FID results on ImageNet 512, alongside deterministic few-step baselines from sCD [Lu & Song, 2025], AYF [Sabour et al., 2025], and DMF [Lee et al., 2025].
>
> | Category | Method | NFE | #Params | FID ↓ |
> |---|---|---:|---:|---:|
> | **Deterministic Few-Step Flow Models** | sCD-L | 1 | 778M | 2.55 |
> |  |  | 2 | 778M | 2.04 |
> |  | AYF-S | 1 | 280M | 3.32 |
> |  |  | 2 | 280M | 1.87 |
> |  | DMF-XL/2+ | 1 | 675M | 2.12 |
> |  |  | 2 | 675M | 1.75 |
> | **Stochastic Few-Step Flow Models** | **MFM-XL/2** | 1 | 683M | 3.37 |
> |  |  | 2 | 683M | 2.63 |
>
> We include a visualization of posterior samples at the **following link**: https://drive.google.com/file/d/10feeJEzpijgMq8hal12S3TtmrefX4bUu/view?usp=sharing, and also report the posterior recovery metric for the ImageNet 512 MFM in comparison with ODE rollouts of GLASS.
>
> | Posterior $t$ | NFE | MFM $\downarrow$ | GLASS $\downarrow$ |
> |---|---:|---:|---:|
> | 0.1 | 1 | 3.293 | 154.837 |
> |  | 2 | 2.364 | 95.944 |
> |  | 4 | 2.147 | 20.238 |
> | 0.5 | 1 | 1.346 | 10.661 |
> |  | 2 | 1.194 | 7.553 |
> |  | 4 | 1.370 | 4.113 |
>
> As with ImageNet 256, MFM remains competitive for unconditional generation and substantially outperforms explicit ODE rollouts for few-step posterior sampling, as reflected in the posterior recovery results across conditioning times and NFE budgets. We will add the full evaluation in the revised version if the paper is accepted.
>
> ### Answer to Question 2:
>
> Thank you for the suggestion. We view MFMs as complementary to RL-style methods, rather than purely as competing alternatives.
>
> Many RL algorithms rely on estimating a value function or advantage. PPO, for example, uses advantages derived from a learned value function, while GRPO estimates advantages from groups of sampled trajectories, effectively yielding a Monte Carlo estimate of the relevant value function. In the diffusion setting, the value function used in such RL-style methods is the same object that appears in our steering/fine-tuning framework. As a result, the expensive SDE rollouts typically used to estimate this value function could instead be replaced by estimates based on MFM one-step posterior samples. In this sense, MFMs are compatible with a broad class of RL-style fine-tuning methods, and provide a potential avenue for significantly reducing their computational cost.
>
> We agree that this is an important direction for future work. At present, we have not run experiments comparing the base RL algorithms, their MFM-augmented variants, and MFM-FT, so we do not want to make empirical claims beyond what we have evaluated. Due to the rebuttal timeline, we are not able to complete these experiments during the discussion period. However, we will clarify this compatibility in the revision and aim to include experiments of this kind in the final camera-ready version.
>
> We hope that our responses address the reviewer’s main concerns, and we kindly ask them to consider increasing their score if they agree.

---

### Decision · Program_Chairs · 2026-04-30

**Decision:**

Accept (regular)

**Comment:**

The consensus among reviewers is that this is a technically solid and well-written paper. The novel "meta" capability of the posterior sampler, while methodologically relying on a fairly straightforward extension of existing flow-map work, offers a principled and scalable path for reward alignment. Despite some minor concerns regarding the scope of experimental baselines and training complexity, the reviewers agreed that the theoretical contributions and the demonstrated efficiency gains make it a valuable addition to the conference. I recommend a solid accept.